# Recent Advances in Electrochemical Sensing Strategies for Food Allergen Detection

**DOI:** 10.3390/bios12070503

**Published:** 2022-07-09

**Authors:** Antonella Curulli

**Affiliations:** Consiglio Nazionale delle Ricerche (CNR), Istituto per lo Studio dei Materiali Nanostrutturati (ISMN), 00161 Rome, Italy; antonella.curulli@cnr.it

**Keywords:** allergens, electrochemical (bio)sensors, nanomaterials, food safety, immunosensors, aptasensors, cell-based biosensors, genosensors, molecularly imprinted polymer (MIP) based biosensors, bacteriophage-based biosensors

## Abstract

Food allergy has been indicated as the most frequent adverse reaction to food ingredients over the past few years. Since the only way to avoid the occurrence of allergic phenomena is to eliminate allergenic foods, it is essential to have complete and accurate information on the components of foodstuff. In this framework, it is mandatory and crucial to provide fast, cost-effective, affordable, and reliable analysis methods for the screening of specific allergen content in food products. This review reports the research advancements concerning food allergen detection, involving electrochemical biosensors. It focuses on the sensing strategies evidencing different types of recognition elements such as antibodies, nucleic acids, and cells, among others, the nanomaterial role, the several electrochemical techniques involved and last, but not least, the ad hoc electrodic surface modification approaches. Moreover, a selection of the most recent electrochemical sensors for allergen detection are reported and critically analyzed in terms of the sensors’ analytical performances. Finally, advantages, limitations, and potentialities for practical applications of electrochemical biosensors for allergens are discussed.

## 1. Introduction

Abnormal reactions linked to food consumption are, in general terms, defined as “adverse reactions to food”. They are defined by the European Academy of Allergology and Clinical Immunology as toxic and non-toxic reactions according to the response mechanism [1]. Toxic reactions trigger identical harmful effects for all individuals and in some cases even poisoning immediately after eating food and they are considered dose dependent. Non-toxic reactions are related to the individual predisposition, are not commonly dose dependent, and are classified as immunological (food allergy) and non-immunological (food intolerance) [2,3].

Food allergy is an adverse immunological response occurring in a reproducible manner after ingestion/exposure to a food component or ingredient. The immune response is described as immunoglobulin E (IgE)-mediated, non-IgE-mediated, or a combination of both. IgE-mediated food allergy is due to the interaction of allergenic proteins with specific IgEs associated with mast cells/basophils present in the intestine and they are the most common. On the other hand, non-IgE-mediated food allergy is ruled by T cells. T cells, also called T lymphocytes, are a type of leukocyte (white blood cell) and an essential part of the immune system. In particular, T-cell response to allergens is associated with regulation of other antibody isotypes such as IgG, IgM, and IgA [4,5].

Food intolerances are adverse reactions to food not involving the immunological system but producing effects comparable to those of a real allergy. In some cases, food intolerances involve an organic pathophysiological process. For example, lactose intolerance is manifested because of the lack or deficiency of the enzyme able to break the lactose molecule. Other foods that can induce intolerance reactions include caffeine in beverages, and tyramine or other vasoactive amines found in cheeses. Finally, some food intolerances cannot be easily explained through well understood organic pathophysiological processes, such as irritable bowel syndrome (IBS), of which the mechanism of production of symptoms is not clear [6].

Since food allergens represent a major food safety concern in industrialized countries and since the only way to avoid the occurrence of allergic phenomena is to eliminate allergenic foods from the diet, waiting for an effective pharmacological treatment, it is mandatory and crucial to have complete and accurate information of ingredients on food labels. Indeed, countries and international bodies are issuing laws, regulations, and standards for labeling of foods, with a complete indication of the allergenic ingredients. Currently, over 200 foods are identified as allergenic, but it must be stressed that some differences in regulations regarding the number of foods indicated as allergenic are evident among different countries.

The European Union has established mandatory labeling regulations for 14 allergenic foods, i.e., eggs, milk, peanuts, nuts, gluten-containing cereals, lupin, soybeans, celery, mustard, sesame seeds, fish, crustaceans, mollusks, and sulfites: therefore, labeling them on their food derivatives is required [7].

On the other hand, in the United States of America, the Food Allergen Labelling and Consumer Protection Act of 2004 [8], effective from 2006, requires a clear indication on the label of the presence of any of the eight major allergenic foods, i.e., milk, eggs, fish, crustacean shellfish, tree nuts, peanuts, wheat, or soybeans. These, also known as the “big eight”, are considered responsible for 90% of all food allergies. According to this law, other allergenic foods are not required to be declared.

In Japan, the indication on the label of the presence of allergenic foods is defined as mandatory or recommended according to the number of cases of actual illness and the degree of seriousness [9]. Consequently, indication of the presence of allergenic foods such as egg, milk, wheat, buckwheat, peanut, shrimp, and crab on the label is mandatory: on the other hand, indication on the label of the presence of the following food products is recommended: abalone, squid, salmon roe, orange, cashew nuts, kiwi fruit, beef, walnut, sesame, salmon, mackerel, soybean, chicken, banana, pork, matsutake, peach, yam, apple, and gelatin.

Moreover, 10 μg allergen protein/g (or mL) food was stated as the threshold value to regulate commercial prepackaged foods; and official food allergen analytical methods, for detecting the threshold value accurately and precisely, were developed and defined.

Also considering the legislation of other countries such as Canada, Switzerland, Hong Kong, New Zealand, and Australia, it is clear that the number of allergenic foods to be indicated on the label is not the same for all, but ranges from a minimum of 5 to a maximum of 14 [10,11]. In addition, no regulatory threshold exists for allergens in food samples, partially justified by the lack of analytical standard protocols. In fact, only Japan indicates a threshold of 10 μg allergen protein/g (or mL) food and which are the official methods of analysis for the validation of the obtained analytical data. In this framework, it is mandatory and crucial to have fast, cost-effective, and reliable analysis methods for the screening of specific allergen content in food products.

As a final comment, although food labeling is required for providing consumers with accurate composition information, accidental ingestion/exposure to some allergens can occur. This exposure can be due to undeclared allergens through adulteration, cross-contamination, or even fraud.

This review is focused on the most recent strategies in electrochemical biosensing for allergen detection. In the literature, several recent and accurate reviews described all the methodological approaches for allergen detection, including biosensors in general and the electrochemical ones in particular and comparing the conventional analytical methods with those more innovative [10,12,13,14,15,16]. Concerning the electrochemical biosensors, two recent reviews reported surveys on the electrochemical biosensing approaches for food safety [17,18], including examples of allergen detection.

Finally, regarding the electrochemical biosensors for allergens, accurate surveys are included in interesting reviews [19,20,21], where different examples were discussed and compared with the conventional analytical approaches.

This review aims to provide an up-to-date survey of the electrochemical biosensors for allergen detection, evidencing the sensing strategies, the role of the electrodic materials and one of the nanomaterials, the type of recognition element involved, and what are the actual advantages over conventional and non-conventional analytical methods.

## 2. Overview of the Conventional Methods for Food Allergen Detection

The determination of allergens in food is complicated because of the small amount of analyte, the complexity of the matrix to be examined, the possible and unexpected changes in the chemical, physical, and immunological properties of allergens due to thermal treatments occurring during food processing, affecting and/or altering their allergenicity [12,20].

General speaking, appropriate detection limits (LODs) for allergens should be between 1 and 100 mg kg^−1^ of food, according to the sample analyzed, but lower detection limits are suggested for highly allergenic foods such as peanuts.

In addition to the effects of food processing, the presence must be considered in some foods containing different allergens that can complicate the determination of the target analyte [12].

The methods reported for the determination of allergens can be divided: immunological, DNA-based, and chromatographic ones. Alternatively, they can be classified as direct or indirect methods, indicating whether the allergens or their biomarkers can be detected.

Immunological methods are based on binding between specific and high affinity antibodies and the epitopes on the target allergen. The epitope, also called antigenic determinant, is portion of a foreign protein, or antigen, capable of stimulating an immune response. Antibodies can be either polyclonal or monoclonal, depending on their ability to bind more than one epitope [10,12,20].

Enzyme-linked immunosorbent assay (ELISA) is the most common immunological method used for the quantitative determination of allergens because of its sensitivity (1–25 ppm), precision, accuracy, easy handling, and standardization capability.

A large number of commercial ELISA kits are available and for a wide selection of food allergens [10]. As disadvantages, ELISA is a time-demanding approach (up to 3.5 h), is expensive if a small number of samples is involved, and the presence of cross-reactions is often reported. In addition, the analytical results are not fully comparable because different immunoreagents are employed [12]. Finally, instrumentation portability and miniaturization are difficult to achieve and produce. Other immunological methods such as lateral flow (LF) assay, dipsticks, rocket immunoelectrophoresis, and dot-immunoblotting must be mentioned, but they are less frequently used. They are rapid and sensitive, but generally are semi-quantitative or at least only qualitative methods. Although less frequently used [20], other methods involved after separation of the allergens are gel electrophoresis, capillary electrophoresis, or high-performance liquid chromatography (HPLC).

DNA methods are based on allergen coding genes and involve the removal of a specific allergen or marker protein encoding a DNA fragment followed by amplification using polymerase chain reaction (PCR) technologies [12]. The main advantages of this method are related to the greater stability of DNA fragments compared to proteins. In fact, the proteins can be denatured both during food processing and/or during the allergen extraction process. The PCR-based methods can allow qualitative, semi-quantitative (PCR-ELISA), or quantitative real-time PCR (RT-PCR) analyses and are suitable for multianalyte detection. However, they do not seem appropriate for food allergens containing a large amount of protein and a low amount of DNA, such as eggs, and the results can be questionable, because processing may affect DNA and proteins in a different way. These PCR methods are considered to be used as additional tools of the immunological methods [20].

Mass spectrometry and other chromatographic techniques have also been used for the identification and/or characterization of allergens in different food products [12,20].

Although still not well diffused for routine analysis, biosensors can be considered as innovative, sensitive, selective, and less expensive, in some cases capable of real-time measurement, environmentally friendly, reusable, and fast: they can be assumed as effective tools for replacing the conventional methodologies previously reported. Both optical and electrochemical biosensors have shown to be appropriate for allergen detection. However, electrochemical biosensors can represent a particularly effective alternative with the proper requirements for on-site, fast, and low-cost analyses, possibly in a real and complex matrix by unskilled personnel.

## 3. Electrochemical Biosensors

Starting from the IUPAC recommended definition, a biosensor is assumed as “an integrated receptor-transducer device, which is capable of providing selective quantitative or semi-quantitative analytical information using a biological recognition element” and in particular “an electrochemical biosensor is a self-contained integrated device, which is capable of providing specific quantitative or semi-quantitative analytical information using a biological recognition element (biochemical receptor) which is retained in direct spatial contact with an electrochemical transduction element” [22] or more briefly as recently reported in the literature is defined as an “electrochemical sensor that has a biological recognition element” [23]. Electrochemical biosensors can be divided into two main categories based on the nature of the biological recognition process, i.e., biocatalytic sensors and affinity biosensors [24,25,26]. Biocatalytic sensors incorporate enzymes, whole cells or tissue slices recognizing the target analyte and produce electroactive species.

According to the literature definition [26], “the term affinity biosensor refers to a device incorporating immobilized biological receptor molecules that can reversibly detect receptor-ligand interactions with a high differential selectivity and in a non-destructive fashion”. It is clear that this definition does not include the biocatalytic sensors.

In other words, affinity biosensors present affinity-based biorecognition elements (BREs) immobilized on or near the transducer. BREs can reversibly and selectively interact with target analytes [25,26]. The principle of affinity biosensors is related to the formation of a stable and selective binding between an appropriate analyte and BREs, producing a response signal through a transducer. BREs incorporated in affinity biosensors are commonly antibody–antigen, oligonucleotides, aptamers, phages, molecularly imprinted polymers (MIPs), and peptide nucleic acids (PNAs). Concerning the electrochemical biosensors for allergen detection, only affinity biosensors are considered and reported in this review.

### 3.1. Electrochemical Biosensor Detection Techniques

Electrochemistry provides a repertoire of very diverse analytical techniques characterized by robustness, low cost, easy handling, and possibility of application to the food safety area [27].

In general, an electrochemical reaction can produce different measurable data, according to the measurable electrical signal generated and consequently the electrochemical technique adopted. In fact, a measurable current can be obtained, and in this case, the corresponding electrochemical techniques are the amperometric ones. Alternatively, a potential can be measured and/or controlled, and in this case, the corresponding electrochemical techniques are the potentiometric ones. Finally, the electrochemical techniques, involving measurements of impedance at the electrode/solution interface, are included in the electrochemical impedance spectroscopy (EIS) method [28,29,30].

Firstly, constant potential amperometry (CPA) is introduced. It is an electrochemical technique in which a fixed potential is applied to the working electrode, and the current generated from the redox reaction is measured. This potential value is generally evaluated by means of other electrochemical techniques such as cyclic and/or linear voltammetry. Under the Faraday’s law rule, the potential applied controls the electrons transfer from and to the analytes, and the measured current is related to the target molecule concentration.

Chronoamperometry (CA) is another type of amperometric technique, in which steady-state current is recorded as a time function by applying a potential to the working electrode. Changes in the current are dependent on the diffusion layer at the electrodes. Near the electrodic surface, the analyte concentration decreases, due to the redox reaction and the diffusion layer controls the analytes approaching the electrode from the bulk solution, inducing a concentration gradient nearby the electrode surface. Chronoamperometry is linked to Cottrell’s equation, establishing the current–time dependence under linear diffusion control at a planar electrode and it is useful to determine the concentration of the analyte once its identity is known using other techniques, such as voltammetry, chromatography, and/or other separation techniques.

Considering the voltametric approach, the current produced from an electrochemical reaction is measured whilst varying the potential window. Since there are many ways to vary the potential, different voltametric techniques are reported. Among others, the most commonly employed are the following: cyclic voltammetry (CV), linear sweep voltammetry (LSV), differential pulse voltammetry (DPV), and square wave voltammetry (SWV).

CV and LSV are widely employed voltametric techniques used to study the electrochemical behavior of an electroactive molecule.

DPV and SWV can be classified as pulse voltametric techniques. They can be used to study the redox properties of extremely small amounts of electroactive compounds for several reasons, but principally: (1) in these measurements, the effect of the charging current can be minimized, so higher sensitivity is achieved and (2) only Faradaic current is measured, so electrode reactions can be analyzed more precisely. All the above-mentioned techniques have been widely employed in the development of electrochemical sensors for different application fields.

A very particular role involves Electrochemical Impedance Spectroscopy (EIS), for what concerns the determination of allergens. Generally speaking, EIS is an effective technique for detecting the interaction between the electrode surface and the analyte by testing the electrode/electrolyte interface and following the change in the impedance or capacitance of the electrode/solution interface. For evaluating the experimental results, a comparison with a theoretical equivalent electrical circuit is required.

Summarizing, EIS represents a powerful approach for investigating and analyzing the interfacial properties related to biorecognition occurring at the electrode surface, such as, for instance, antibody–antigen recognition or substrate–enzyme interaction. There are two main EIS approaches used in biosensing applications: Faradaic and non-Faradaic.

The conventional EIS or Faradaic approach involves electrolytes containing redox species undergoing electrochemical reactions. In this case, the signal of conventional EIS transducers is mainly due to impedance changes at the electrode electrolyte interface, easily monitored by means of the charge transfer resistance (Rut).

In a non-Faradaic approach, there is theoretically no electron transfer and the changes in the capacitance double layer are detected. In fact, the proximity of the double layer to the electrode surface can be used to detect the interaction between analyte and the probe surface functionalized with BREs such as aptamers, anti-bodies, and so on. In other words, the signal of non-Faradaic EIS transducers is mainly due to capacitance changes at the electrode electrolyte interface, easily monitored by means of the double-layer capacitance (Cdl). This approach is well known as Electrochemical Capacitance Spectroscopy (ECS) and is widely used as a transduction approach for biosensing because of its high sensitivity. The principle of capacitive biosensors is based on the change in the electrical surface or thickness of the dielectric layer on the electrode surface. The working electrode modified with receptor molecules presents a stable capacitance, and the consequent interaction with a target analyte will produce variations in capacitance.

In particular, the non-Faradaic technique can be applied not only to the analyses in the presence of dielectric films, but in developing sensors with improved analytical performance.

In order to deepen the topic, several particularly meaningful reviews are suggested, including interesting examples of impedimetric biosensors [31,32,33,34,35,36].

Finally, we would like to introduce a relatively new non-conventional EIS approach: Dynamic Electrochemical Impedance Spectroscopy (DEIS).

It can be assumed as a combination between EIS measurement with the traditional CV, where high frequency impedance spectra are measured while the potential is scanned to simultaneously carry out CV. In the case of DEIS, the charge transfer resistance depends on the applied potential program, i.e., on the scan rate. In this way, the performances of CV and EIS are paired advantageously, limiting, for example, the surface contamination, and making the EIS measurements faster. It should be underlined that there are few examples of applications in the sensing area and that there is still a long way to go. To get more information about DEIS, there are several articles and reviews in the literature clarifying the theoretical aspects of this approach [37,38,39,40,41,42,43].

### 3.2. Electrodes, Sensing Materials, and Devices

A variety of electrode materials ranging from noble metals to carbon including also conductive polymers is available for several biosensing applications, in particular, for allergen detection the most common electrodic materials are gold and carbon.

The peculiar properties of gold (Au) (biocompatibility, stability, and conductivity) have supported its use as electrodes in biosensors. The gold electrode sensitivity and functionality can be improved by modifying its surface, introducing suitable molecules, polymers, and nanomaterials. In addition, gold is a material particularly suitable for micro-fabrication and for the immobilization of biomolecules [18,44].

Carbon-based electrodes include a spectrum of materials ranging from graphite to the well-known glassy carbon (GC). GC is the most used electrodic material in biosensing, probably because of its particular properties such as conductivity, mechanical strength, regenerability, and large potential window, among others. On the other hand, the requirement of GC’s accurate pretreatment procedure before use, can condition its application in the electrochemical sensing area [18].

Nanomaterials and nanostructures play a significant role in designing and improving the electrochemical biosensor performances introducing several peculiar functionalities on the electrodic surface and taking advantage from their high surface-to-volume ratio [18,44,45]. As a general comment, nanomaterials and nanostructures can be assumed to integrate at the nanoscale level the properties of the macroscopic electrodes.

Nanoparticles are the most common examples of 0-D nanomaterials, i.e., they represent nanomaterials with all the dimensions in the nanoscale. Gold nanoparticles were widely used, because the corresponding synthesis processes are well established and easier to perform, in addition to their biocompatibility, stability, conductivity, and their high surface-to-volume ratio [44,45].

Magnetic nanoparticles (MNPs) are another 0-D nanomaterial employed in electrochemical biosensors, in particular, for efficient separation after the application of an external magnetic field [44,46]. On the other hand, the magnetic particles, or beads (MBs), with micrometric dimensions are commercially available and can be used as effective separators before biosensing, even if MNPs seem to be more appropriate for the application to the miniaturized devices owing to their higher surface-area-to-volume ratios, greater stability in suspension, and less predisposition for the agglomeration and/or aggregation in the presence of a magnetic field.

Nanotubes, nanorods, and nanofibers are defined as 1-D nanomaterials, i.e., nanomaterials with only two dimensions in the nanoscale. In this review, the examples reported in Section 4 include carbon nanotubes (CNTs) and carbon nanofibers (CNFs).

CNTs present several properties associated with their structure, functionality, morphology, and flexibility and can be classified as single-walled carbon nanotubes (SWCNTs), double-walled carbon nanotubes (DWCNTs), and multi-walled carbon nanotubes (MWCNTs) depending on the number of graphite layers. The chemical functionalities for their application in biosensing can easily be performed through the tubular structure modification [45], for promoting the electron transfer between BREs and the electrodic surface.

CNFs have gained attention in the biosensing area because of their peculiar electrical conductivity, comparable to that of conducting polymer nanofibers. CNFs should not be confused with CNTs, as CNFs have cylindrical nanostructures with graphene layers organized in different shapes.

Even if CNTs have higher electrical conductivity than CNFs, the structure of graphene sheets in CNFs provides significant advantages, making CNFs more appealing than CNTs for the electrochemical biosensing field, since an enhancement of electron transfer with respect to that of CNTs is promoted by the stacking of CNF graphene sheets. In addition, the surface of CNFs can be activated without damaging their structure [44].

Graphene and its derivatives can be considered as 2-D nanomaterial because they have only one dimension in the nanoscale, and they include typically plate-like shapes.

Graphene shows properties such as high conductivity, accelerating electron transfer, and a large surface area, very similar indeed to the corresponding properties of CNTS, so it is considered a good candidate for assembling sensors to determine several analytes. Different graphene-based materials have been produced (e.g., electrochemically, and chemically modified graphene) using many procedures and for more details different reviews are available in the literature [44,47].

Finally, we consider hybrid nanomaterials: they can be assumed as a synergistic blending of nanomaterials with other nanoscale materials and/or polymers, resulting in a new nanomaterial, which not only improves the properties of the starting materials but can also provide their own peculiar features.

Combining different typologies of nanomaterials and/or nanostructure make it possible to assemble high-performance electrochemical biosensors, taking advantage from the combined nanomaterial properties [44,45]. Hybrid nanomaterials have been widely used in different ways as transducers, signal amplifiers, and labels in electrochemical biosensors. In Section 4, examples of the electrochemical biosensors for allergen detection exploiting different hybrid nanocomposites will be reported and discussed.

Following the miniaturization trend towards portable instrumentation for on-site detection, simple and low-cost biosensing platforms including screen-printed electrodes (SPEs) have received increasing attention and interest instead of conventional platforms, involving more conventional laboratory equipment [48,49]. Screen-printed electrodes (SPEs) are produced at the industrial level by depositing a combination of layers onto a flat substrate by means of a printer [49] and an accurate and recent review concerning all the production steps of SPEs is available in the literature [50]. SPEs are produced in a wide range of geometries, with different materials, can be modified and mass produced, and they are cost-effective [49,51]. Their reproducibility, robustness, and stability have contributed to their widespread diffusion. SPEs coupled with electrochemical techniques have represented and represent a new generation of miniaturized biosensing platforms, accelerating the transition from conventional benchtop instrumentation to low-cost, robust, and portable sensing devices.

Laser-scribed graphene electrodes (LSGEs) represent an evolution of the concept of SPEs. LSG can be considered as a new electrode material and is assumed as a three-dimensional (3D) graphene, i.e., a 3D mesoporous network with high conductivity and electrocatalytic activity. It can be printed using a mask-less process from precursor materials such as carbon, polymers, or biopolymers, checking and optimizing the morphology, composition, and deposition method. For more details and information, a recent review is available in the literature [52].

Finally, in this review, examples of electrochemical microfluidic devices for allergen analysis are reported and discussed. It must be underlined that advances in technology have significantly improved the implementation of microfluidic devices (μFDs). For this reason, they are considered unique analytical tools in several application areas ranging from diagnostic to food safety. Low reagent consumption, small sample volume, low cost, shorter analysis times, possibility of multiplex screening, and mainly the miniaturization of an entire laboratory in a single chip (lab-on-chip, LOC)) represent the most remarkable advantages, concerning the application of μFDs [44,53].

More recently, the utilization of paper as an alternative to traditional microfluidics materials and its potential for analytical applications has attracted particular and wide attention.

The interest in paper for analytical applications can be explained with low cost, fine thickness, low-weight and flexibility, compatibility with different patterning methods, disposability, and easy functionalization with proper groups. The development of paper-based μFDs can be clarified accounting for the potentiality of these devices to perform analyses commonly carried out by means of benchtop instrumentations. In particular, electrochemical paper-based analytical devices (ePADs) have shown high sensitivity and selectivity, owing to a proper selection of electrode materials, electrochemical technique, and/or recognition elements. Significant trends in research on ePADs include studies on electrode fabrication, electrochemistry at the electrodes on paper, strategies to improve analyte detection, and potential applications of ePADs, and recent literature reviews examine and deepen these features [54,55,56,57,58,59].

Last, but not least, we would like to introduce Origami μPADs (OμPADs). Origami is the traditional Japanese art of paper folding, and it is a 400-year-old technique for creating 3D geometries starting from a single piece of paper [60,61]. An origami-based method addresses the issues linked to a multilevel μPAD design. OμPADs are patterned on a single sheet of chromatographic paper and then folded into a 3D fluidic architecture using simple principles of origami (that is, no adhesive tape or scissors are allowed). In this way, the sampling area is connected to the electrode by folding the paper.

Several interesting and recent reviews are available in the literature, concerning this topic [54,55,56,59,62,63].

## 4. Electrochemical Biosensors for Food Allergen Detection

The electrochemical biosensors represent a valid alternative for the determination of allergens in food, combining the sensitivity of electrochemical transducers with the high specificity of recognition processes. As previously indicated at the end of the Introduction Section, herein, affinity electrochemical biosensors for allergen detection are considered. They are mainly classified as antibody-based biosensors (immunosensors), aptamer-based biosensors (aptasensors), nucleic-acid-based biosensors (genosensors), and cell-based biosensors, depending on the different biological recognition molecules immobilized on the electrode surface. Molecularly imprinted polymer-based biosensors are included even if the recognition element is not properly a biological molecule but is assumed as a synthetic recognition element acting as a biological receptor. Finally, we would like to introduce bacteriophage-based biosensors, where chemically and thermally stable virus nanoparticles serve as the biorecognition elements.

In the following subsections the main characteristics of each type of biosensors are reported and discussed, together with significant examples of application to allergen detection.

In this review, we considered the detection of the most common allergens contained in milk (α-lactoglobulin, β-lactoglobulin, casein), wheat (gliadin), in shrimp (tropomyosin), in egg (lysozyme, ovalbumin, ovomucoid), in tomato (Sola l 7), in mustard (Sin a 1), in kidney beans (lectin), in soy (β-conglycinin, glycinin), and in peanuts and hazelnuts (Ara h 1, Ara h 2, Ara h 6, Cor a 14, Cor a 9).

Concerning the electrochemical biosensors for milk allergens and lysozyme detection, we want to suggest two recent and interesting reviews [64,65], presenting the development of immunosensors and aptasensors and analyzing weaknesses and strengths and future challenges.

Finally, we would like to point out that as far as the determination of allergens is concerned, there is only one example of an electrochemical sensor, unlike what was reported in a previous review regarding the electrochemical biosensors for food safety [18].

Abaci [66] developed an electrochemical sensor modifying a pencil graphite electrode with graphene oxide (GO) to determine β-lactoglobulin (β-LB). It is well known that β-LB, the main whey protein, causes an allergic reaction in humans and it is one of the main reasons for cow milk allergies [64,67,68].

Considering the reported sensor, the bioreceptor was substituted by a nanomaterial mimicking the bioreceptor action and activity. In this way, the issues linked to the immobilization protocols can be avoided, but accurate studies and analyses of the toxicity and degradation of the nanomaterials are suggested.

It is well known that hydrogen peroxide (H_2_O_2_) is widely employed as an antibacterial agent in milk, so monitoring H_2_O_2_ is crucial to follow the transformation of milk in cheese and to detect by-products from the enzymatic reactions [66]. Graphene oxide (GO)-modified graphite electrodes are extensively used as non-enzymatic H_2_O_2_ sensors because the nanomaterial acted increasing the electrode active surface area, the electrocatalytic activity, and accelerating the electron transfer from and to the electrode.

The PGE electrode was modified by dipping in a GO suspension for an appropriate time period (50 min). The operating principle of the sensor was based on the fact that H_2_O_2_ produces hydroxyl radicals such as ·OH and ·OOH at working redox potential when reacting with β-LB [66] and a decrease in the corresponding electrochemical response is observed. In other words, the H_2_O_2_ current signal resulted as inversely proportional to the β-LB amount, i.e., the higher the β-LB concentration, the lower the H_2_O_2_ electrochemical response.

A linear concentration range of 0.53–11.16 mg mL^−1^ with a detection limit of 0.27 mg mL^−1^ was evidenced. The sensor was applied to spiked milk samples, obtaining recoveries between 90.00 and 118.30%. Moreover, the analytical results were comparable with those acquired with ELISA and UHPLC as reference external methods. Unfortunately, the sensor selectivity, reproducibility, repeatability, and stability were not investigated.

### 4.1. Immunosensors

A large number of electrochemical biosensors for allergens reported in the literature are immunosensors. Generally, antibodies are immobilized on the electrodic surface and the operating principle of the electrochemical immunosensors is based on the conversion of the results of the immunochemical reaction among the antibodies and the target allergen molecules into an electrochemical signal proportional to the target concentration.

The performance of an electrochemical immunosensor is strictly correlated to the immobilization method of biorecognition elements, which should ensure the stability of the antibodies on the transducer surface, maintaining their specificity and biological activity [69,70]. Strategies for the immobilization of antibodies were recently reviewed [69,70]. The adsorption including electrostatic, hydrophobic, and van der Waals interactions is considered attractive and easy to perform. However, the immobilized antibodies were randomly oriented, reducing antigen-binding capacity, and desorption can occur compromising the immunosensor stability and reproducibility. Therefore, this strategy is not commonly applied. Another approach is the covalent immobilization based on the interactions of the functionalized transducer surface with proper functional groups of the antibodies. This immobilizing method can be carried out via a cross-linker such as glutaraldehyde (GA) or via a covalent binding involving 1-ethyl-3-(3-dimethylaminopropyl)carbodiimide (EDC) and N-hydroxysuccinimide (NHS) and it can improve the sensor stability and reproducibility, but the issue of the antibody orientation is not completely solved.

The immobilization of the BRE is usually followed by the incubation of the immunosensor with blocking agents, such as bovine serum albumin (BSA), casein, and surfactants for preventing non-specific adsorption on the antibodies or on the electrode surface. This non-specific adsorption can imply a decrease in the sensor sensitivity, being crucial for its application in real complex matrices.

Let us now examine the immunoassay design. Electrochemical immunosensors can be classified following the classical immunoassay design, i.e., label-free, sandwich (noncompetitive), and competitive immunosensors. Label-free electrochemical immunosensors are easy to assemble, present a fast response, and the possibility of real-time monitoring. Antibodies and antigens are not commonly electrochemically active; for this reason, a redox probe is added to the solution. The formation of the antibody–antigen immunocomplex stops the electron transfer between the electrode and the redox probe, changing the analytical signal. For example, a decrease in the peak current while increasing the target concentration is observed, using a voltametric technique. In contrast, an increase in resistance to charge transfer is observed as the concentration of the analyte increases, if EIS is used as an electrochemical analysis technique.

In sandwich-type design, after the immunochemical reaction between the biorecognition element (primary antibody) and the target, a labeled secondary antibody is introduced.

Consequently, the formation of the sandwich complex produces an electrochemical signal proportional to the concentration of the analyte.

The labels used for assembling sandwich-type immunosensors are of course electroactive compounds such as enzymes and electrocatalysts. It must be considered that the introduction of labels on the secondary antibody can produce a biosensor complex structure and can enhance related costs but on the other hand, improves the biosensor’s performance if compared to the label-free format.

In competitive electrochemical immunosensors, labeled and free biomolecules compete for the binding sites, present on the electrodic surface.

It is to be underlined that the competitive approach is preferred when the detection of small molecules is involved. In fact, small molecules are not suitable for the sandwich assays or for the label-free strategies; for this reason, this format is not involved for allergen detection.

As a first example of an immunosensor, we would like to introduce a label-free immunosensor for the determination of α-lactoglobulin (α-LB), based on the detection of α-LB via the α-LB antibody (α-LB-Ab) entrapped in a polypyrrole (PPy) film [71]. Briefly, regarding α-LB, it is a protein present both in human and cow milk and it is recognized to have beneficial effects on child development. For this reason, α-LB, coming from cow milk, is added in commercial infant milk formulas to make them the most similar to breast milk. It is important to have fast and reliable methods for determining α-LB, especially considering its allergenic potential. The proposed sensor is based on a gold screen-printed electrode where the α-LB-Ab buffered solution mixed with a pyrrole (PY) solution was drop-casted.

Then, electropolymerization was carried out for entrapping the antibody. An efficient antibody immobilization resulted because of the electrostatic interaction between the positively charged polypyrrole chains with the negatively charged carboxyl groups of α-LB-Ab antibodies.

After the optimization of α-LB incubation time and the corresponding incubation temperature, α-LB was electrochemically detected via DPV, obtaining a linearity concentration range from 355 to 2840 pg mL^−1^ and a limit of detection (LOD) of 0.19 fg mL^−1^. The sensor selectivity was analyzed substituting first the target molecule with a nonspecific target such as albumin from human serum (HSA) and then the antibody using IgG as the non-specific antibody. In addition, the detection of α-LB was performed without antibody entrapment on the electrode surface, as a control experiment. The results showed that the specificity and selectivity of an immunosensor is correlated to the use of α-LB-Ab and the immobilization strategy. Unfortunately, the sensor’s reproducibility, repeatability, and stability were not investigated.

The immunosensor was applied to detect α-LB in real spiked samples of different types of milk (UHT whole milk, low-fat milk, dry milk, and almond milk) with a recovery ranging between 93 and 97%, but a comparison of the results with an external standard method was not provided.

Coming back to β-LB detection, already introduced at the end of the previous section, we consider an electrochemical immunosensor based on screen-printed carbon electrodes (SPCEs) modified by a simple drip coating using a nanocomposite. The nanocomposite (PEI-rGO-AuNCs) included reduced graphene oxide (rGO) functionalized with polyethyleneimine (PEI) and gold nanoclusters (AuNCs) modified with glutathione (GSH) [72]. A β-LB antibody (β-LB-Ab) was then immobilized on the nanocomposite, inducing a reduction in SPE conductivity and the current change due to the immunoreaction reaction between antigen and antibody was recorded for the β-LB detection.

The immunosensor design involved the assembly of PEI-rGO-AuNCs on the electrode surface, as illustrated in Figure 1. It has to be evidenced that PEI-rGO-AuNCs is stabilized because of the electrostatic interactions between PEI-rGO, positively charged for the presence of –NH_2_ groups of PEI and AuNCs, and negatively charged for the presence of GSH. Therefore, the integration of AuNCs and PEI-rGO provided a nanohybrid with enhanced electrical conductivity and a higher active surface area.

The morphological analysis revealed a large number of AuNCs uniformly distributed on the PEI-rGO surface. The immunosensor performances were investigated by means of DPV, taking into account that the electrochemical response decreased as the target concentration increased, because of the reduction in SPCEs conductivity due to the immunochemical reaction.

The sensor showed an LOD of 0.08 ng mL^−1^ and a detection range from 0.01 to 100 ng mL^−1^ for β-LB. The sensor reproducibility was acceptable (RSD% 1.9%).

Ovalbumin (OVA), bovine serum albumin (BSA), egg lysozyme, and casein were regarded as possible interfering proteins for testing the sensor selectivity, but the electrochemical response was not affected by their presence.

Finally, the sensor’s long-term stability was studied and after two weeks in the refrigerator at 4 °C, the electrochemical response showed a decrease of only 4.3%.

Furthermore, milk-spiked samples from four milk brands were analyzed, and the results agreed with those from ELISA, but no recovery data were provided.

The Ybarra group [73] proposed an electrochemical biosensor based on a sandwich-type immunoassay for the detection of β-LB, using screen-printed carbon nanotube electrodes. The electrodes were printed using a carbon nanotube ink modified with polystyrene beads (PSBs) bearing several carboxylic groups for the bioreceptor immobilization. This strategy showed interesting sensing performance if compared to those obtained using CNTs functionalized by means of oxidative treatments. The primary antibody was immobilized onto the electrode surface by means of EDC reaction among the carboxylic groups of PSBs and the –NH_2_ groups of the primary antibody and horse radish peroxidase (HRP) was the label for the secondary antibody. Briefly, the sandwich immunoassay involved the primary antibody on the electrode surface reacting with an allergen and its presence was evidenced by the secondary antibody, labeled with HRP. An LOD of 0.173 ppm was achieved and β-LB was detected in the range from sub ppm level to 10 ppm.

Since this immunosensor was developed for the detection of β-LB in rinse samples after cleaning of production lines, proteins that could contaminate the surface of the equipment were selected and tested for cross-reaction. BSA, casein, and soymilk extract were evaluated, and no significant cross-reaction was found, except a casein cross-reactivity of 1%. Unfortunately, the sensor’s reproducibility, repeatability, and stability were not investigated.

The next examples are focused on the design and assembly of biosensors for detecting the protein gliadin, responsible for a wide variety of gluten-related disorders, such as celiac disease. The first example involved a gliadin label-free type immunosensor [74]. The anti-gliadin polyclonal antibodies were trapped on the surface of GCE modified with a collagen coating via cross-linking promoted by transglutaminase (TG) [75]. The gliadin is then detected by specific immunoreaction with anti-gliadin polyclonal antibodies. Since the immunoreaction is expected to significantly change the interfacial properties of the electrode/solution interface, EIS is used as the detection technique. The sensor showed an LOD of 5 mg L^−1^ and a detection range from 5 to 20 mgL^−1^. Concerning the sensor selectivity, casein and soy proteins were analyzed as possible interfering agents, but the immunoreaction was specific only for gliadin. It must be underlined that the stability, reproducibility, and repeatability of this sensor were not investigated, and it was not applied to real samples, so no comparison with data coming from an external reference method was provided.

The next example involved a sandwich-type immunosensor where a SPCE is modified with carbon nanofibers (CNFs) and then it was connected with an immunosensing paper platform [76]. CNFs on SPCE acted to enhance the electrochemical active area and they were functionalized via acidic treatment to improve their solubility and dispersion, exactly as the functionalization of CNTs is performed. The morphological analysis of the modified electrode surface showed a homogeneous CNFs film with a compressed 3D structure.

The paper surface presents hydroxyl groups available for the immobilization of the antibodies, but being inactive in pure cellulose, their activation by plasma oxidation is required for promoting the covalent bonding of antibodies on the paper surface. This treatment created several aldehyde groups, able to form Schiff bases with the amino groups of the antibodies [77]. The paper platform was placed on the CNFs/SPCE, after the anti-gliadin antibody immobilization, the incubation of the analyte, and the successive addition of gliadin antibody labeled with HRP, as usual. The electrochemical determination was carried out by means of amperometry, obtaining a linearity range from 0 to 80 μg kg^−1^ and an LOD of 0.005 mg kg^−1^. The selectivity, reproducibility, and stability were investigated. Albumin, casein, glutenin from wheat, β-LB, and folic acid were tested as possible interfering compounds, using the same concentration of gliadin. No significant cross-reaction was found except for casein. Consequently, BSA was preferred to casein as the blocking agent. The sensor precision was analyzed with intra- and inter-assay approaches, obtaining for the intra-assay an RSD% ranging from 3.87 to 5.13% and for the inter-assay an RSD% ranging from 5.23 to 6.56%.

The sensor stability was investigated storing CNFs/SPCEs and the immunosensing paper platform at 4 °C in the refrigerator for three months. After this period, no appreciable decrease in the electrochemical response was evidenced.

Finally, the sensor was applied to real samples of flour (manioc flour, rice flour, gluten-free flour, and common wheat flour). The relative recoveries ranged from 98.50% to 102.10% with an RSD% less than 4.93%. Finally, these results were comparable with those obtained with ELISA as the reference external method.

Pirvu and co-workers [78] developed a label-free immunosensor based on TiO_2_ nanotubes (TiO_2_ NTs), GO, and gliadin antibodies. It is well known that TiO_2_ NTs were used in sensor applications because they can be easily prepared with high reproducibility and low cost, they have a large surface area, and they are non-toxic, hydrophilic, and biocompatible [78,79]. These nanostructures also have other important characteristics: they have antibacterial activities and confer UV protection [80,81]. TiO_2_ NTs were prepared electrochemically by anodization of a Ti electrode, followed by a thermal treatment for increasing their crystallinity. TiO_2_ NTs/GO composite was prepared by electrodepositing GO onto the nanotubes. The gliadin antibody was then immobilized using EDC/NHS protocol, after the electrode surface functionalization with pyrene carboxylic acid allowing a covalent bond with the gliadin antibody. The role of TiO_2_ NTs was to improve the electrochemical active surface area and GO acted to enhance the conductivity and the electron transfer to and from the electrode surface. From the morphological analysis of the electrode surface, it was evident that the NT walls were thicker after the GO electrodeposition and the immobilized antibody partially covered the TiO_2_NTs/GO nanocomposite. The gliadin was detected by EIS, obtaining an LOD of 14 ppm and a limit of quantification (LOQ) of 45 ppm, to be improved because in the literature it is reported that a food can be labeled as “gluten-free” if it has a gluten content below 20 ppm. A linear concentration range from 0 to 20 ppm was achieved. Unfortunately, no data concerning the stability, the reproducibility, the repeatability, and the possible applicability to real samples of the sensor were provided.

Many egg-white proteins are known to be allergenic. Ovomucoid (OM, Gal d 1) and ovalbumin (OVA, Gal d 2) represent the most important allergens.

An interesting sandwich-type immunosensor for OM detection was described by the Pingarron group [82]. This approach includes the sandwiching of OM involving allergen antibodies immobilized onto carboxylic-acid-functionalized magnetic beads (HOOC-MBs) and HRP-labeled allergen antibody. The functionalization of MBs with carboxylic groups supports the OM antibody immobilization on them via the EDC-NHS protocol. It is well known in the literature that MBs acted to minimize the matrix effect and improve the sensitivity and the analysis time [83,84]. The resulting magnetic immunocomplexes were captured on the surface of SPCE to perform the amperometric detection.

After the optimization of the experimental conditions, a linearity range from 0.3 to 25 ng mL^−1^, an LOD of 0.1 ng mL^−1^, and an LOQ of 0.3 ng mL^−1^ were obtained. The reproducibility was considered acceptable in terms of RSD% (6.0%) and the stability was investigated storing the OM antibody immobilized onto MBs in buffer at 4 °C for 63 days and checking every day the immunosensor analytical performance. After 63 days, no significant decrease in the response signal was evidenced. Concerning the selectivity, conalbumin, ovalbumin, lysozyme, avidin, and riboflavin, being proteins present in egg whites, were tested as possible interferences, but no cross-reaction was detected.

The immunosensor was applied to spiked egg-white samples as well as to spiked wheat flour and bread samples, obtaining results in accordance with literature data (egg-white samples) [85] and lower than those coming from the ELISA method (wheat flour and bread) [86].

A label-free electrochemical immunosensor for the detection of OVA was developed using a nanocomposite based on iron oxide and palladium nanoparticles (Fe_3_O_4_@PdNPs) and a natural polymer chitosan (CHI) for modifying a screen-printed graphene electrode (SPGE) [87]. Fe_3_O_4_@PdNPs were prepared by chemically reducing K_2_PdCl_6_, the PdNPs precursor, onto Fe_3_O_4_ nanoparticles and then they were dispersed in a CHI suspension. The nanocomposite suspension was casted on SPGE and 4-amminobenzoic acid (4-ABA) was electrografted onto the modified SPGE to assist the antibody immobilization and improve the electron transfer. The OVA antibody was immobilized via EDC/NHS protocol, using the –COOH groups of 4-ABA. Under optimized experimental conditions, OVA was detected by DPV and a linear concentration range of 0.01 pg mL^−1^–1 μg mL^−1^ with an LOD of 0.01 pg mL^−1^ was achieved. A comparison of the immunosensor analytical performances with the ELISA method was performed, evidencing a better sensitivity, probably due the presence of the nanocomposite and to the functionalization of 4-ABA.

The sensor reproducibility was studied with interesting results in terms of RSD% (0.28%).

The long-term stability was investigated keeping the sensor for 20 days at 3 °C and the signal response showed a decrease of only 3.6%, after this period. BSA, lysozyme, and casein were tested as interferences and no significant changes or decreases in the electrochemical response were observed.

The immunosensor was applied to spiked real food product samples, with recoveries ranging from 101.6 to 107.0%.

It is well known that OVA is used for the clarification of wines, promoting tannin removal, together with convalbumin and ovomucoid, fish collagen, and horse gelatin, among others [88]. However, OVA traces in wine can trigger allergic reactions in particularly sensitive subjects.

A disposable electrochemical microfluidic device (DEμD) based on a sandwich immunosensing platform was developed for the detection of OVA in wine samples [89]. In fact, the sensing platform involves the sandwiching of OVA including OVA-polyclonal antibody and HRP-labeled OVA polyclonal antibody immobilized on MBs, as reported for OM [82]. The DEμD assembly involves the use of eight SPCEs as working electrodes (8-WEs). These electrodes were modified with a bilayer assembled by means of the electrostatic interaction between a polycation such as poly (diallyldimethylammonium chloride) (PDDA) and GO. The OVA polyclonal antibody was immobilized via the EDC/NHS protocol on GO/PDDA/8-WEs and the HRP-labeled polyclonal antibody was immobilized on MBs (OVA-HRP-Ab-MBs), functionalized with –COOH groups as reported for OM [82]. The immunocomplex between OVA and OVA-HRP-Ab-MBs antibody, produced by the immunoreaction of the analyte and the HPRP-labeled antibody, was injected in DEμD where the immune-sandwich on 8-WEs was generated and the electrochemical response for OVA detection was investigated by amperometry. More details on the injection system and the corresponding analytical procedure are available in [89]. Under optimized experimental conditions, a linear concentrations range of 0.01–10 pg mL^−1^ and an LOD of 0.2 fg mL^−1^ were obtained.

The repeatability of the DEμD was evaluated and the RSD% values were 5.9% using the same DEμD and 7.0% with three different DEμDs.

The stability of the OVA Ab on 8-WEs was also considered. For this reason, different arrays were prepared on the same day and stored at 4 °C in buffer. Different microfluidic devices were tested on different days for the OVA detection. The DEμD devices showed a decrease of 5.1%, in the amperometric responses, after three days. After storage of 5 and 10 days, the electrochemical responses of the immunosensor were reduced by 14% and 29%, respectively, indicating that the OVA Ab immobilization on 8-WEs was rather stable. The stability could be improved if the electrodes were stored in dry conditions. The DEμD was applied to spiked real samples of white and red wines and the results are comparable with those coming from the ELISA protocol.

Several hazelnut proteins are considered as allergens and among them, Cor a 14 (2S albumin) can cause serious allergic reactions probably thanks to its difficult digestion and thermal stability. Consequently, it cannot be degraded or deteriorated during the heat treatments taking place during food processing [90]. 2S albumins are the most important class of seed storage proteins widely distributed in cotyledonous plants [91]. As storage proteins, they support the plant as a nutrient source during its growth, but, unfortunately, some 2S albumins are classified as food allergens.

Recently, two label-free electrochemical immunosensors were developed for determining Cor a 14, using two types of customized antibodies, namely anti-Cor a 14 IgG (raised in rabbit) and anti-Cor a 14 IgY (raised in hen eggs) [92]. The antibodies were immobilized via EDC/NHS protocol on AuSPEs, after the self-assembling monolayer (SAM) functionalization of the Au electrode surface with mercaptosuccinic acid (MSA). After the immobilization, the morphological analysis showed an electrode surface with a spherical/globular type structure with a smooth surface profile, indicating an antibody cross-linking on the modified surface, decreasing the electrode roughness. After the optimization of the experimental conditions and parameters, the electrochemical detection of Cor a 14 was carried out by means of SWV and the results obtained using the two antibodies were similar in terms of linear concentration range (0.1 fg mL^−1^–0.01 ng mL^−1^). However, the anti-Cor a 14 IgY such as BRE seems to show better analytical performance and in particular an LOD of 0.05 fg mL^−1^, this is probably due to its greater affinity for the allergen, but the LOD related to the other immunosensor was not provided.

The specificity of both anti-Cor a 14 IgY (raised in hen eggs) and anti-Cor a 14 IgG (raised in rabbit) was tested against 2S albumins from peanut and other tree nuts, in particular, cashew nut, chestnut, almond, pecan nut, macadamia, walnut, and Brazil nut. The anti-Cor a 14 IgY (hen egg) presented higher affinity to the target allergen Cor a 14 and at the same time, it was much less reactive towards the other 2S albumins, with the exception of the 2S albumin from peanut (Ara h 2) and walnut (Jug r 1), probably due to their structural similarity and to their similar epitopes. The repeatability of the two immunosensors was evaluated with acceptable results in terms of RSD% (<5%). Stability and reproducibility data of the immunosensors were not given. The anti-Cor a 14 IgY, raised in avian species, evidenced a better specificity and sensitivity, maybe due to phylogenetic biodiversity with respect to the antibody raised in a mammalian species, in fact rabbit and human are both mammals.

The electrochemical immunosensor with anti-Cor a 14 IgY was applied to samples of wheat with different % of hazelnut protein as models of a real complex matrix. The results indicated that the immunosensor can determine 0.16 mg kg^−1^ of hazelnut protein in wheat. This means that the proposed method is able to detect traces of Cor a 14 in foods; thus, resulting very effective for protecting the hazelnut-allergic population [93].

Peanut is one of the principal allergenic foods, containing potentially allergenic proteins such as 7S globulin or vicilin (Ara h 1), 2S albumin (Ara h 2), and 11S globulin or legumin (Ara h 3). Vicilins such as 2S albumins and legumins are seed storage proteins particularly abundant in legumes and tree nuts (representing about 20% of their protein content depending on the species). They are recognized as thermostable and resistant to digestion in the human body [90,94]. A nanodiamond-based voltametric sandwich-immunosensing platform was developed for the Ara h 1 detection in peanuts [95]. The nanodiamonds (NDs) were drop-casted onto an SPCE and then the capture antibody was immobilized on the modified electrode. The sandwiching involved Ara h 1, capture antibody immobilized on NDsn and streptavidin-alkaline phosphatase (S-AP)-labeled secondary antibody. A scheme of the sensor assembly and the immunosensing mechanism is reported in Figure 2.

Under optimized experimental conditions, a linearity range of 25–500 ng mL^−1^ with an LOD of 0.78 ng mL^−1^ were achieved. The reproducibility and repeatability were investigated with satisfactory results in terms of RSD% (7.3 and 4.9%, respectively). The storage stability was addressed and the immunosensor was stable for two weeks in a moist environment at 2–8 °C. Ara h 2, Ara h 6, and OVA were selected as interfering molecules, but the electrochemical response of Ara h 1 was not affected by the presence of non-specific allergens. The immunosensor was applied to spiked real samples of biscuits, crackers, cookies, cereals, energetic/protein bars, and the results were comparable with those provided by the producers. Finally, it was validated with the ELISA standard method.

We would like to introduce the crustacean allergies, and tropomyosin (TPM) has been considered as the most serious shellfish allergen. TPM is a muscle protein with a regulatory function, acting together with the troponin complex [96].

A sandwich format amperometric immunosensor has been developed including magnetic nanoparticles (MNPs) and SPCEs for the detection of shrimp TPM [97].

The synthesized MNPs were provided with the appropriate carboxylic groups for the covalent binding of the antibodies through –NH_2_ coupling via EDC/NHS protocol. MNPs offer higher surface-area-to-volume ratios, greater stability in suspension, and less predisposition for the agglomeration and/or aggregation in the presence of a magnetic field with respect to the commercial MBs, as already reported in Section 3.2.

As usual, the immunosensor involved a sandwiching of TPM including TPM antibody and HRP-labeled TPM antibody immobilized on MNPs. A homemade poly (methyl methacrylate) (PMMA) dock with an encapsulated permanent magnet was employed to capture the magnetic particles onto the WE surface. Under optimized experimental conditions, a linearity range of 0–218.7 ng mL^−1^ and an LOD of 46.9 pg mL^−1^ were obtained. The LOD value is four times lower than that obtained from the ELISA method [97]. The selectivity of the immunosensors was investigated analyzing and comparing the electrochemical responses using TPM coming from pork, chicken, beef, crab sticks, and squid. It must be underlined that the immunosensor was able to discriminate among TPM with different origins, evidencing a decrease in the signal responses ranging from 87 to 93% with respect to that of shrimp TPM.

A nanocomposite based on gold-microrods (AuMRs), Pd-nanoparticles (PdNPs), and polyaniline (PANI) was employed to modify an SPCE and to assemble a label-free electrochemical immunosensor for the shrimp TPM detection [98]. Commercial AuMRs were casted on the electrode surface, then PdNPs were electrodeposited onto them, and finally the aniline electropolymerization was carried out, as illustrated in Figure 3.

AuMRs, PdNPs, and PANI acted together to improve the conductivity, to accelerate the electron transfer, and to enhance the sensor stability. The TPM antibody was immobilized via EDC/NHS protocol and under optimized experimental conditions, TPM concentrations between 0.01 pg mL^−1^ and 100 pg mL^−1^ were investigated by means of DPV, with a detection limit of 0.01 pg mL^−1^.

OVA, BSA, Casein, and lysozyme were tested as possible interfering proteins and they did not affect the electrochemical response of TPM. The sensor reproducibility was satisfactory with an RSD% of 3.96% and the sensor stored at 4 °C in the refrigerator was stable for six days. Finally, the immunosensor was applied to spiked real samples of shrimp-free cream crackers with recoveries ranging from 84.1 to 117.6% and RSD% ranging from 1.3 to 10.3%. Comparisons with data coming from an external reference method were not given.

Kidney bean lectins (KBLs) are proteins that are not degraded during food processing heating treatments and can trigger adverse reactions [99]. Very few methods have been developed for a rapid detection and monitoring of lectin in kidney-bean-derived foodstuffs [99]. A label-free voltametric immunosensor for the direct determination of KBL has been proposed based on a gold nanoparticle–polyethyleneimine–MWCNTs nanocomposite (AuNPs/PEI-MWCNTs) [100]. The nanohybrid was synthesized by one-pot procedure for enhancing the electrochemical response. In particular, PEI acted both as a dispersing agent to avoid the agglomeration/aggregation of the nanotubes and as an in situ reducing agent of the AuNPs precursor. The nanocomposite was casted onto a GCE, and a recombinant Staphylococcal protein A (SPA) functionalized with cysteine (CYS) was immobilized on it through the interaction of the CYS thiol group with the AuNP surface to provide an appropriate platform for an oriented KBL polyclonal antibody immobilization. SPA-mediated oriented antibody immobilization has been applied to the immunosensor assembly [101], improving the biosensor analytical performance owing to a better interaction between the antibody and target analyte [102]. After the analytical parameter optimization, the electrochemical detection of KBL was carried out by means of DPV, with a linearity range of 0.05–100 μgmL^−1^ and an LOD of 0.023 μgmL^−1^. The selectivity was analyzed using black turtle bean lectin, concanavalin A (Con A), BSA, and γ-globulin. No clear interference has been detected except in the case of black turtle bean lectin, but the two proteins are very similar in the amino acid sequences (98.1%). The reproducibility study gave acceptable results in terms of RSD% (2.24%). The long-term stability was investigated storing the immunosensor at 4 °C, observing a signal decrease of <10% after 4 days, of 13.55%, and of 28.64% after 8 and 15 days, respectively. Finally, the immunosensor was applied to spiked real samples of raw and cooked kidney bean milks with recoveries ranging from 90.96 to 97.18% and the results were comparable to those coming from the ELISA conventional method.

Mustard is one of the most important spices causing allergy because of its wide diffusion and its high allergenic degree [103]. Three allergens, i.e., Sin a 1, Bra j 1, and a 11s globulin have been identified from mustard seeds. Sin a 1 and Bra j are classified as 2S Albumins (see also Cor a 14), usually called napins and found in dicotyledonous seed [90,91]. Sin a 1 is assumed as the main allergenic protein and marker for the mustard allergy diagnosis. It is heat resistant and slightly affected by food processing exactly as Cor a 14. The first electrochemical immunosensors for Sin a 1 detection have been developed by the Pingarron group [104]. It involved a sandwich immunoassay where a capture antibody and a detector antibody, labeled as usual with HRP were included. MBs were used for immobilizing the capture antibody and after the sandwiching among the capture antibody, target protein, and detector antibody, the immunocomplex was magnetically immobilized onto SPCEs, incorporated in a PMMA dock as previously reported [97]. The electrochemical Sin a 1 determination was amperometrically performed and under optimized experimental conditions a linearity range of 2.7–50 ng mL^−1^ was obtained with an LOD of 0.82 ng mL^−1^ (0.82 ppb). The reproducibility was acceptable in terms of RSD% (6.3%), using eight immunoplatforms, prepared in the same way and on the same day. Concerning the stability, the MBs modified with capture antibody were kept at 4° C in sterilized buffer and every day for 50 days the modified MBS were used to assemble the immunoplatforms for the detection of Sin a 1. No significant differences in the signal responses were found.

The immunosensor selectivity was investigated and a 2S albumin was used as possible interference such as Pin p 1 from pine nut, but the electrochemical signal was not significantly affected. In addition, the selectivity was tested considering the raw extracts from different plants containing different 2S Albumins such as pine nut (Pin p 1), peanut (Ara h 2, Ara h 6), rape seed (Bra n 1), cashew (Ana o 3), and yellow mustard (Sin a 1) and in this case the immunoplatforms were selective, showing a clear response only for the yellow mustard (Sin a 1). The magnetoimmunoplatform analytical data on the raw plant extracts were comparable with those coming from the ELISA conventional method.

Soybean (*Glycine max*) is a good source of high-quality proteins, fibers, and essential fatty acids, as well as vitamins, minerals, and so on, but also of allergenic proteins; thus. as already mentioned in the Introduction Section, soy and its derivatives are listed as allergenic food. Glycinin and β-conglycinin are the most abundant proteins in soybean, classified as storage globulins [105]. They are considered as the main allergenic proteins and markers for soy allergy diagnosis. The Pingarron group developed a sandwich-type immunosensing platform, using specific antibodies for glycinin and β-conglycinin and carboxylic-acid-modified MBs [106]. The sandwich immunoassay involved a capture antibody and a detector antibody, labeled as usual with HRP, as illustrated in Figure 4.

MBs were used for immobilizing the capture antibody and exactly as for the Sin a 1 immunoassay, after the sandwiching among the capture antibody (cAb), target protein, and detector antibody, the immunocomplex was magnetically immobilized onto SPCEs, incorporated in a PMMA dock [97]. After the optimization of the experimental procedures, considering the detection of the allergens using two different immunosensors, a linearity range of 0.1–125 ng mL^−1^ for β-conglycinin and of 0.1–100 ng mL^−1^ for glycinin were obtained with an LOD of 0.03. ng mL^−1^ for β-conglycinin and of 0.02 ng mL^−1^ for glycinin. The reproducibility data were acceptable with an RSD% of 3.8 and 3.7% for β-conglycinin and glycinin, using five platforms prepared in the same way. The storage stability of the cAb-MBs bioconjugates was checked by monitoring the amperometric responses obtained with the bioplatforms prepared using the stored bioconjugates in the absence and in the presence of β-conglycinin and glycinin. The electrochemical responses were comparable for at least 42 days after the bioconjugate preparation.

The two immunoplatforms were applied to spiked real samples of raw cookie dough and baked cookies enriched with soy flour, with recoveries of 101% for glycinin and ranging from 93 to 99% for β-conglycinin. The results were also validated with the ELISA reference method. Dual (SPdCE) screen-printed carbon electrodes were modified to detect the two allergens at the same time. The analytical performances are comparable with those coming from SPCEs, also including real samples analysis, only the sensitivity slightly decreased, maybe due to a smaller active area of SPdCE with respect to that of SPCE.

As a consequence of the fact that allergens may show cross-reactions, it is necessary to improve multiplex analytical systems for the detection of different allergens using a single sample, so reducing the analysis time and costs and assuring consumers about the content of food, highlighting the possible presence of allergens. Another factor to consider is the possibility of using portable user-friendly devices that can allow analysis, for example, at the restaurant or at home [13,107]. In this framework, the integrated exogenous antigen testing (iEAT) is a very interesting example of a user-friendly and simple smartphone-based electrochemical food analyzer based on a sandwich immunomagnetic assay format [108] in analogy with the examples of immunosensors reported above [73,74,82,89,97]. The target proteins were gliadin in wheat, Ara h 1 in peanut, Cor a 1 in hazelnut, casein in milk, and OVA in egg white. The iEAT device comprises a disposable extraction kit, with extraction buffers and wash solutions, a multichannel electrode, a customized potentiostat, plugged through a Bluetooth connection to a smartphone for controlling the system and uploading data to a cloud server. Summarizing, the system includes a disposable allergen extraction device and an electronic keychain reader for sensing and communication, as shown in Figure 5.

The extraction kit captures and concentrates target proteins from food products. The captured allergens are then electrochemically and quantitatively determined by means of chronoamperometry. Overall, the iEAT system enables a fast, accurate, and cost-effective quantitative allergen detection. Considering the consumer-friendly aspect, the extraction kit is simple to use, and the integrated communication protocols allow users to record and upload data in a cloud server. The iEAT detection showed very interesting results in terms of LOD, i.e., for gliadin 0.075 mg kg^−1^, for Ara h 1 0.007 mg kg^−1^, for Cor a 1 0.089 mg kg^−1^, for casein 0.170 mg kg^−1^, and for OVA 0.003 mg kg^−1^ and these data are comparable with those coming from the ELISA standard method. The iEAT assay was applied to real food products, starting from packaged food (bread, milk, cereal) and desserts (cookies, ice cream). Next, foods coming from restaurants such as burgers, pizza, dressed salads, and beers were investigated. As a general comment, it is to be evidenced that gluten-free or nut-free foods are properly free of gliadin and Ara h 1, respectively, but unexpected allergens could be detected such as gluten in nut-free cookies. Moving to foods from restaurants, the data showed that some allergens were detected as expected such as gliadin in hamburgers, but unexpected allergens were found such as gliadin in dressed salads, probably coming from the dressing, OVA, and casein in beers. In fact, OVA is used as an additive for wine clarification [88] and for improving beer foam quality [108] and casein as a stabilizing agent for beer [108]. However, the production of toxic waste coming from the analytical protocol does not allow to consider the portable device completely environmentally and consumer-friendly because of lack of appropriate assessment of the waste disposal.

As a conclusive comment regarding the reported examples of immunosensors for allergen detection, we can observe that the LODs, independently of the analyte, achieved ng mL^−1^ or fg mL^−1^ in several examples. Concerning the immunosensor format, it is not possible to indicate a preferred format, the choice between label-free and sandwich seems to be equivalent. Furthermore, with regard to the label-free format, the problem of being antibodies-oriented does not seem to be taken into account except in one case [100].

Regarding the immunosensors using the sandwich format, the presence of MBs or even MNPs seems to greatly improve the performances of the sensors.

Questionable points are represented by data relating to the selectivity, applicability to real samples, and subsequent validation with an external method; in fact, these issues are not always adequately addressed. The analytical performance of the reported immunosensors for the determination of allergens as well as the corresponding sensor formats are summarized in Table 1.

### 4.2. Aptasensors

Aptamers (from the Latin word aptus (“fit”) and from the Greek word meros (“part”)), are essentially short and single-stranded DNA or RNA oligonucleotides, containing almost 20–80 nucleotides and with a molecular weight ranging from 6 to 30 kDa. They are able to interact specifically with a target molecule, similarly to the antigen–antibody interaction and consequently the aptamers are generally assumed as chemical antibodies [109,110]. The aptamers are selected from a large collection and assortment of nucleic acids, providing roughly 10^15^ different sequences, using the well-known Systematic Evolution of Ligands by Exponential enrichment (SELEX) technology [109,110]. Aptamers can make strong and specific interactions with a broad range of targets such as proteins, nucleotides, antibiotics, toxins, cancer cells, viruses, bacteria, and allergens. These interactions include van der Waals forces, hydrogen bonding, and electrostatic interactions among others, mimicking the antigen–antibodies binding mechanism. Even if until now antibodies are considered as the most reliable biorecognition elements, the aptamers are beginning to replace them, thanks to their facility of synthesis and chemical functionalization, structural homogeneity among different lots, and a wide collection of targets. In addition, aptamers can specifically recognize and bind their target molecules, even if their molecular weight is lower than that of antibodies but is resulted sufficient for a specific biorecognition action. Despite the advantages provided by aptamers, including their thermal and pH stability and their low immunogenicity, their use is still limited although growing. Briefly, immunogenicity is defined as the ability of a biological system such as antigens, aptamers, cells, and/or tissues to cause an immune response and is generally considered to be an undesirable physiological response.

A possible explanation of the aptamers’ restricted use may be linked to the fact that their production must be placed on an industrial scale at lower costs and that the sharing of inter-laboratory studies must be increased; thus, improving the quality and reliability of the synthesized aptamers.

Electrochemical aptasensors can be classified according to the adopted detection approach: labeled type where labels (antibodies, enzymes, metal nanoparticles, or redox compounds) are covalently or non-covalently linked to aptamers, or label-free type [109]. In both cases, the electrochemical response is correlated to the target concentration. As for immunosensors, sandwich format was recently introduced, combining an aptamer (the capture aptamer) with the target on the sensor or electrodic surface and then binding another aptamer (signaling/secondary aptamer); thus, increasing the sensitivity of the detection system. It is evident that the capture aptamers are immobilized on the transducer and secondary signaling aptamers are used for signal measurement. Another strategy involves redox probes covalently linked to the aptamers. The recognition mechanism includes some conformational modifications in the aptamer structure or shift in the target binding strand, so producing redox current changes. In other words, a signal-on electrochemical aptasensor showed a current increase with the target concentration, while a signal-off aptasensor showed a current decrease as the target concentration increased. Finally, analyzing a competitive approach, a competition takes place between a free oligonucleotide and the target molecule for specifically binding an aptamer immobilized on the electrode.

The immobilization procedure represents a crucial step in establishing the aptasensor’s performance, so guaranteeing significant reactivity, proper orientation, specific access, and stability of the aptamer-modified surface as well as minimizing unexpected non-specific adsorption phenomena [64]. The selection of the immobilization approach is correlated on the assay format and affects the aptamer’s capability to bind the target protein. The most diffused approaches are the covalent bond, the affinity reaction, and the self-assembled layer. Different molecules such as tetra (ethylene glycol) (TEG), for example, acting as arm spacers were introduced to provide proper stability, to preserve the surface coverage and to save the same binding affinity as in solution.

The immobilization of the aptamer is usually followed by the incubation with appropriate blocking agents, so limiting a possible decrease in the sensor sensitivity due to non-specific adsorptions on the aptamer or on the electrode surface, as already mentioned in Section 4.1 for the immunosensors.

As a first example, we would like to describe the electrochemical label-free aptasensor for β-LB developed by Eissa and coworkers [111]. The selected aptamer BLG14 showed good affinity and specificity vs. β-LB A and B proteins, the most common cow’s milk β-LB variants [112]. It was immobilized on a commercial graphene screen-printed electrode (GSPE) through a stable physical absorption due to the π-π stacking interactions between the aptamer and the graphene layer.

The electrochemical SWV signal of [Fe(CN)_6_]^3−^/^4−^ as a redox probe was monitored before and after the aptamer immobilization and after the interaction aptamer/protein. A decrease was observed just after the aptamer adsorption (signal-off) because of the repulsion between the redox probe and the aptamer, both negatively charged. After the interaction of aptamer-β-LB, the redox probe signal increased, suggesting a more efficient electron transfer, probably due to a change in the aptamer conformation, inducing its release from the electrode surface, followed by a decrease in negative charge onto the surface (signal-on). The scheme of the aptasensor assembly and detection mechanism is presented in Figure 6.

Consequently, this electrochemical response increase was used for the allergen detection. A linear concentration range of 100 pg mL^−1^–100 ng mL^−1^ and an LOD of 20 pg mL^−1^ were achieved. The LOD resulted lower than those obtained by HPLC and ELISA methods, as reported in the literature [112]. BSA was tested for the aptasensor selectivity, and no significant response was evidenced. In addition, the sensor stability was analyzed and after a storage of a week, no electrochemical response decrease was found.

Finally, the aptasensor applicability to spiked real food samples such cake, cheese crackers, and biscuits was examined, evidencing acceptable results in terms of recoveries ranging from 90 to 95%. Unfortunately, reproducibility and repeatability data were not available.

A disposable electrochemical platform based on poly(aniline-co-anthranilic acid) (PANI/PAA) copolymer coupled with an aptamer to detect β-lactoglobulin was developed by the Marrazza group [113]. PANI/PAA film was electrodeposited on the graphite screen-printed electrode (SPGE) surface and the copolymer modified electrode was used for covalently immobilizing an aptamer via the EDC/NHS protocol. An oriented and flexible covalent immobilization of aptamer was obtained via –COOH functionalities of the copolymer. It must be underlined that the aptasensor used a competitive assay with a signal-off mechanism. In fact, the hybridization reaction between the immobilized aptamer and its biotinylated complementary sequence was performed.

When the β-LB protein was absent, the aptamer, hybridized with its biotinylated complementary sequence, presented a rigid duplex structure, labeled with streptavidin–alkaline–phosphatase conjugate. After addition of the enzymatic substrate such as 1-naphthylphosphate, the enzyme catalyzed the hydrolysis of 1-naphthyl-phosphate to 1-naphthol, detected by DPV. While, as β-LB was present, the duplex structure including β-LB-aptamer and its complementary sequence was disabled. The current peak height decreased with increasing β-LB concentration, decreasing the 1-naphthol amount coming from the enzymatic reaction. After optimizing the experimental parameters, a dose–response curve was obtained between the 0.01 and 1.0 μg mL^−1^ β-LB concentration range with a limit of detection of 0.053 μg L^−1^. Casein and BSA at the concentrations usually present in milk samples [113] were used for evaluating the aptasensor selectivity with acceptable results. The stability was further evaluated, storing, and checking 30 aptasensors in dry conditions at 4 °C for one month and they were stable.

Finally, the aptasensor was applied to spiked real samples of soy and cow milk with recoveries ranging from 80 to 85% (soy) and of 95% (cow). A comparison with data coming from an external standard method were not given.

Another sensing approach always used to determine β-LB in food samples was designed by the Marrazza group [114] using a folding-based electrochemical aptasensor based on graphite screen-printed electrodes (GSPEs), modified with AuNPs/poly-L-lysine nanocomposite, and a β-lactoglobulin aptamer labeled with methylene blue (MB) as the redox probe. This aptamer changed its conformation in the presence of β-LB, and, therefore, the space between MB and the electrode surface decreased and the electron transfer was enhanced. The AuNPs/poly(-L-lysine) nanocomposite improved the electroactive area and the thiolated aptamer was immobilized on AuNPs via covalent bonding. If β-LB was not present, there was more distance between the redox probe (MB) and the electrode surface, so the signal is lower, but in the presence of the target protein, the aptamer changed its conformation, reducing the distance between MB and the electrode surface and increasing the corresponding electrochemical response. Under optimized experimental conditions, the electrochemical detection of β-LB was carried out by DPV. The aptasensor response was linear for concentrations of β-LB within the range 0.1–10 ng mL^−1^, with a limit of detection of 0.09 ng mL^−1^, lower or at least comparable with those reported in the literature by means of conventional methods [115,116]. The reproducibility was analyzed obtaining acceptable results in terms of RSD% (RSD% < 13%). The selectivity was tested against casein and the β-LB electrochemical response was not affected by the presence of the interfering molecule. The stability was further evaluated, storing, and checking aptasensors in dry conditions at 4 °C and they resulted stable for several weeks, but the exact number of the weeks was not indicated.

Finally, the aptasensor was applied to spiked real samples of biscuits and yogurt, with recoveries ranging from 103 to 117% (biscuits) and from 95 to 116% (yogurt).

An interesting example of an aptasensor for β-LB detection was developed by the Huang group [117], including the DNA aptamer, Au@flower-like bismuth vanadate (Au@BiVO_4_) microspheres, and AuNPs. In particular, the DNA aptasensor was prepared through the sequential adsorption of DNA1 oligonucleotide and a β-LB aptamer on an indium tin oxide electrode (ITOE), modified with electrodeposited AuNPs (AuNPs/ITOE).

It is well known in the literature that nucleic acid aptamers have been widely used in biosensors instead of antibodies, because of their high selectivity with their target molecules [118]. Au@BiVO_4_ microspheres evidenced an intrinsic peroxidase mimic catalytic activity and played a significant role in the signal amplification [117]. Summarizing the sensing strategy, the DNA aptamer guaranteed the selectivity for β-LB analysis, AuNPs improved the conductivity of the working ITOE, and Au@BiVO4 microspheres modified with a DNA2 oligonucleotide acted as the labeled probe, as illustrated in Figure 7. The aptamer could bind together with β-LB and consequently, the β-LB-aptamer complex left the AuNPs/ITOE surface. Next, the DNA2/Au/BiVO_4_ probe resulted locked to the DNA1/AuNPs/ITOE through the DNA2/DNA1 hybridization, since DNA1 and DNA2 are complementary sequences. In other words, as β-LB concentration increased, the number of the DNA2/Au/BiVO_4_ probes, hybridized on the surface of the working electrode, raised, consequently to the removal of the target–aptamer complex from the electrode surface, and thereby an enhancement of the electrochemical signal was evidenced.

The electrochemical β-LB was performed by amperometry and a linear concentration range of 0.01–1000 ng mL^−1^ and an LOD of 0.007 ng mL^−1^ were achieved. Casein, BSA, γ-LB, and lactoferrin were tested as possible interfering proteins, but their electrochemical responses were not significant with respect to that of β-LB. The reproducibility was acceptable in terms of RSD% (2.59%). Analyzing the long-term stability, a decrease of only 11.4% in the initial response was observed after 30 days at −20°C in wet conditions, and such a result was assumed as satisfactory. Finally, the sensor applicability to real spiked infant food formula samples was investigated, with recoveries ranging from 92 to 103.5%. In addition, these data resulted comparable to those coming from the ELISA standard method with an RSD% of ≤5.4%.

Recently, the Chen group has developed an aptasensor for β-LB detection based on a tri-functional hairpin (HP) [119]. In particular, HP included an appropriate aptamer part, a nicking site with two complementary sequences named site-1 and site-2 and a DNA sequence (named as T1) for amplification. The aptamer and the nicking enzyme (endonuclease) supported the electrochemical signal amplification in a hybridization chain reaction (HCR) system. Analyzing the sensing strategy, in the absence of the target protein, the aptamer sequence resulted hybridized with T1 so that a stable stem-loop structure was created. It is well known that the stem-loop structure is an intramolecular base pairing occurring in single-stranded DNA or RNA if the sequences of two regions of the same strand are complementary to each other [120]. The stem-loop structure could be opened in the presence of sequences complementary to the loop sequence. This feature can be used for assembling biosensors for detection of target biomolecules such as nucleic acids or proteins such as β-LB. As a general comment, an effective stem-loop structure evidences the maximum opening in the presence of the target sequence but cannot be opened by the non-target sequences. The properly designed stem-loop requires adequate experimental conditions such as buffer type, incubation time, and hybridization temperature for implementing the sensor performance. Coming back to the β-LB aptasensor, the aptamer captured the target protein, so a conformational change in HP was induced and the site-1/site-2 complementary sequences hybridized and T1 oligonucleotide was exposed. The endonuclease could identify the nicking site and T1 could be released and electrochemically detected with the support of the HCR system. In particular, a gold electrode was modified with electrodeposited AuNPs and an HCR system, involving an oligonucleotide S1 and two hybridized hairpins (HP1 and HP2) labeled with methylene blue (MB), were immobilized on the electrode surface. Two sequences of HP1-MB and HP2-MB matched with the T1 sequence, so the released T1 can interact with them forming T1-HP1-MB and T1-HP2-MB rigid structures. In the absence of the T1 release, an initial current could be recorded by means of DPV, but after the T1 conjugation, a current decrease was observed as the β-LB concentration increased. A linearity range from 0.01 to 100 ng mL^−1^ with an LOD of 5.7 pg mL^−1^ was achieved. The repeatability was acceptable with an RSD% of 2.75%, while the reproducibility was analyzed, evaluating the intra- and inter-assays. For the intra-assay, an RSD% ranging from 1.5 to 2.6% and for the inter-assay, an RSD% ranging from 1.7 to 3.5% were obtained. OVA, BSA, egg lysozyme, and casein, as non-target proteins present in milk, were considered for the selectivity tests and their electrochemical responses were not significant with respect to that of β-LB. Considering the long-term stability, after 21 days at 4 °C, an electrochemical response decrease of only 9.3% was observed. The aptasensor was applied to spiked real hypoallergenic formula (HF) milk samples with recoveries ranging from 94.5 to 101.4% and RSD% ranging from 2.19 to 3.62%, but no comparison with data coming from an external method was provided.

As the last example of an aptasensor for β-LB detection, we would like to introduce a label-free non-Faradaic capacitive aptasensor using a Laser Scribed Graphene (LSG) electrode [121]. Laser Scribed Graphene (LSG) is assumed as an innovative approach to synthesizing in situ graphene directly into a desirable electrode pattern on a flexible substrate such as polyimide (PI)/kapton, carbon paper. As already mentioned in Section 3.2, LSG electrodes represent an evolution of the concept of SPEs, and more details are available in a recent literature review [52].

1,1-Carbonyldiimidazole (CDI) is an organic molecule used in the coupling of amino acids for peptide synthesis. It is a highly reactive carboxylating agent with two acyl imidazole leaving groups which can activate carboxylic and hydroxyl groups for conjugating nucleophiles [122]; consequently, CDI acted as a cross-linker with a similar reaction mechanism for carboxylic acids and alcohols. Moreover, the imidazole functional groups of CDI exhibit a strong interaction with the copper atom, forming charge-transfer complexes [123].

CDI was used together with copper to synthesize a nanocomposite (CDI-Cu NFs), including Cu nanoflowers (NFs) and acting as a support for immobilizing the aptamer for β-LB detection. The larger specific surface area of CDI-CuNFs enhanced the electrochemical response through an adsorption of a larger amount of the aptamers. CDI-CuNFs were synthesized according to the literature procedure [121] and casted onto the LSG electrode surface, functionalized with a carboxylic group. Next, the aptamer was immobilized after a surface modification with neutravidin. The scheme of the aptasensor assembly and the β-LB detection is reported in Figure 8.

The electrochemical detection of β-LB was performed by means of non-Faradaic capacitance. This electrochemical technique has been recently adopted in the biosensing area and it was already described and discussed in Section 3.1.

The aptasensor analytical performances were particularly appealing and tricky with a linearity range of 1 ag mL^−1^–100 fg mL^−1^ with an LOD of 1 ag mL^−1^. BSA and lysozyme were tested as milk non-target proteins and their electrochemical response was found not significant with respect to that of β-LB. The reproducibility was investigated using three different aptasensors and interesting results were obtained with RSD% ≤ 9.1%. The stability was assessed considering a storage of seven days and a decrease in response of 26.59% was obtained. The aptasensor was applied to spiked real food samples (Herbalife meal replacement shake from Formula 1) with recoveries ranging from 92.95 to 94.99%, but no comparison with data coming from an external standard method was provided.

Lobo-Castañon and co-workers proposed a competitive electrochemical aptasensor [124] for PWG gliadin analysis, based on the specific targeting of the Gli-4 aptamer through a competition between PWG gliadin or gluten and 33-mer peptide, being the gliadin immunodominant peptide and assumed as the primary initiator of the allergy response to gluten [125]. PWG gliadin is a reference material produced under the guidance of the Prolamin Working Group (PWG), gliadin being the wheat prolamin. Analyzing the sensor assembly, a SPCE was modified immobilizing the 33-mer peptide on a streptavidin layer. A competition between the 33-mer peptide and PWG gliadin for a defined concentration of biotinylated Gli-4 aptamer was introduced. As the PWG gliadin concentration increased, the amount of aptamer bound to the peptide on the electrode surface decreased. The bound aptamer is determined through the detection of the enzymatic product using HRP as the label by chronoamperometry. A correlation between the aptamer bound to the surface and the current measured was found and the chronoamperometric current was inversely proportional to the concentration of the PWG gliadin in the range from 1 to 100 μg L^−1^ with an LOD of 0.113 mg L^−1^. The reproducibility was investigated with RSD% <11%, but, unfortunately, stability, selectivity, and repeatability data were not given. Gluten content was measured with the reported aptasensor in spiked real samples of gluten-free snacks and foods and in rolled oats and the results were comparable with those coming from the official immunoassay performed by two accredited laboratories.

Albanese and co-workers developed a label-free impedimetric aptasensor based on Gli-1 aptamer and poly (amidoamine) dendrimer of fourth generation (PAMAM G4) [126]. Aptamers Gli-1 and Gli-4 presented both high affinity vs. Mer-33 and for this reason are currently used for gliadin detection [124,127,128], but Gli-4, the ligand with the highest affinity, can be inadequate for gliadin detection in hydrolyzed food samples, while Gli-1, the most abundant aptamer among those binding to the peptide, is evaluated as kinetically favored [129]. PAMAM G4 was used for the Au thin-film single-electrodes functionalization via glutaraldehyde cross-linking, so enhancing the aptasensor analytical performance. The aptamer was then immobilized by physical adsorption on Au-modified electrode. The scheme of the biosensor assembly, and the detection of gliadin is reported in Figure 9.

The PWG gliadin detection was monitored by EIS and two linearity ranges of 5–50 and 50–1000 mg L^−1^ with a limit of detection of 5 mg L^−1^ were obtained. The reproducibility was analyzed, and RSD% was lower than 5%. After storage at 4 °C for two months, the electrochemical response presented a minimal decrease. Selectivity and repeatability data were not given. The aptasensor was applied to spiked real samples of beer, gluten-free beer, rice, gluten-free bread, and corn flour and the obtained results were comparable with those coming from the ELISA method.

Recently, the Toniolo group reported a label-free impedimetric aptasensor for gliadin, based on AuNPs as the immobilizing platform and on Gli-4 truncated aptamer (Gli-4T) with high affinity with the target [130]. AuNPs provided not only a stable immobilization for Gli-4T, but, as usual, an improvement in the electron transfer from and to the electrode and in the active surface area [45]. AuNPs were electrodeposited on an SPCE, and a layer of streptavidin was then immobilized by adsorption on the modified electrode. Finally, the biotinylated aptamer was immobilized on modified SPCE. In Figure 10, a scheme of the aptasensor design and sensing strategy is provided.

The redox probe [Fe(CN)_6_]^4^^−/3^^−^ was employed to evaluate the presence or absence of PWG-Gliadin. 

The PWG gliadin was detected by EIS and a linearity range of 0.1–1 mg L^−1^ (corresponding to 0.2–2 mg L^−1^ of gluten) with an LOD of 0.05 mg L^−1^ of gliadin (corresponding to 0.1 mg L^−1^ of gluten) were determined. The electrochemical response showed no significant decrease after five days. Selectivity, reproducibility, and repeatability data were not given. The aptasensor was applied to spiked real samples of gluten-free beer and gluten-free soy sauce, where gluten could be present in the hydrolyzed form and where the Gli-1 aptamer use was suggested. The results were acceptable with recoveries ranging from 93 to 101% and it is to be underlined that probably the use of a truncated aptamer with the highest affinity together with a label-free format allowed to analyze hydrolyzed real samples. In fact, it is fully recognized that the elimination of non-essential nucleotide regions to control the affinity interactions improved aptamer affinity for targets; thus, boosting the targets–aptamer interaction by strengthening aptamer–target complexes [131].

The next two examples concerning the gliadin detection involved the use of a deep eutectic solvent (DES) named ethaline [132].

DESs are classified as ionic liquids with high solubilization power, with improved biodegradability and low toxicity and they were employed in different fields such as electrochemistry and organic synthesis among others. It is evidenced that using DES in electrochemical sensors assembly can improve the corresponding analytical performances, as already reported in the literature [133,134]. Ethaline is a mixture of choline chloride and ethylene glycol, which is a very efficient extraction medium for gluten from both unprocessed and processed food [135], so avoiding and limiting the need for several dilutions of samples, which can affect the sensor analytical performance introducing errors and cross-contaminations.

A sandwich aptasensor for gluten was assembled incorporating two identical biotin-labeled truncated aptamers, one as a capture aptamer immobilized on a carbon screen-printed electrode and the other as a reporting aptamer by binding gliadin after incubation in streptavidin-peroxidase [136]. A comparison between two truncated aptamers such as Gli-4T and Gli-1T was performed, evidencing that Gli4-T improved the aptamer–protein affinity in ethaline, due to the DES stabilizing action on the aptamer. On the contrary, Gli-1T decreased its affinity probably because it was negatively affected by the truncation. This sensor determined PWG gliadin in ethaline by means of chronoamperometry, with a dynamic range between 1 and 100 μg L^−1^, an LOQ of 1 μg L^−1^, and an intra-assay coefficient of variation of 11%. Selectivity was tested using rice flour and soya extracts in ethaline and their electrochemical response was found not significant with respect to that of PWG gliadin, considering that rice and soya are naturally gluten-free. Stability and repeatability data were not given. The aptasensor was applied to real spiked samples of dessert powders, panna cotta and vanilla cream, considered as gluten-free foods. The results were lower than those obtained by the ELISA method performed in an external certified laboratory.

Recently, the Toniolo group developed a paper-based electrochemical sensing platform including aptamer–antibody sandwich and an SPCE for the detection of gluten in a DES such as ethaline [137]. Aptamer–antibody sandwich assay is able to improve sensor sensitivity and specificity, as already reported in the literature [138,139]. Gli4-T aptamer was employed as the capturing element, while a 401/21 antibody was used as the detection probe, labeled with HRP. In Figure 11, a scheme of the aptasensor assembly and the sensing strategy is provided.

The sensor determined gluten, extracted in DES, by chronoamperometry with an LOD of 0.2 mg L^−1^, a dynamic range between 0.2 and 20 mg L^−1^, and an intra-assay coefficient of variation of 10.69%. Stability and repeatability data were not given. The sensor was applied to real samples of gluten-free flour and corn flakes and the data resulted comparable with those obtained from the ELISA method. Finally, the selectivity was investigated comparing the results obtained from the ethaline extracts of chickpea flour, rich in proteins and naturally gluten-free with those obtained from the blanks and the experimental data were comparable.

Lysozyme (muramidase or N-acetylmuramic hydrolase, Lys) is an enzyme present in different organisms playing several essential actions. It is also called “the body’s own antibiotic” because of its antibacterial activity in the human body. Lysozyme has a peculiar role in the food industry as a preservative due to its antibacterial activity. In particular, it is useful in the wine industry, acting as stabilizer and blocking the fermentation instead of sulfites. In addition, it prevents butyric acid bacteria action in cheeses, beer, meat, and shrimps. On the other hand, Lys can trigger allergic reactions in sensitive individuals, even in trace amounts; thus, Lys determination in foods is becoming relevant. Therefore, it is considered as one of the five major allergenic proteins together with OM, OVA, ovotransferrin, and ovomucin in chicken eggs. A review, published in 2021, presents the electrochemical and optical aptasensors for Lys detection in human fluids and in foods, reporting analytical performances, adaptability, and real sample applications [65], while in this review we present the most recent examples of aptasensors for Lys detection in food products.

The first example describes an aptamer-based biosensor for Lys detection, using the ink-jet printing technique for stable and reproducible immobilization of an aptamer [140]. In particular, a dispersed CNT–aptamer complex was prepared and used as a printable ink. In the ink structure and composition, strong π-π stacking interaction between the DNA nucleotide bases of the single-stranded DNA and the CNT sidewalls were involved, and the aptamer amount was managed through the printing layer number. After the deposition of the CNT-aptamer ink, the sensor is then used for the Lys detection using EIS and Fe(CN)_6_]^3−^/^4−^ as the redox probe. Briefly, in the absence of Lys, the aptamer slowed down the redox probe electron-transfer and a large charge resistance (Rct) was evidenced. In the presence of the target, the aptamer bound preferentially to Lys, because its affinity vs. the protein is higher than that vs. CNTs. Consequently, Rct resulted decreased because the redox probe electron transfer increased. Under optimized conditions, a linear concentration range from 0 to 1.0 μg mL^−1^, with an LOD of 90 ng mL^−1^, was achieved. After 21 days at room temperature, the aptasensor response resulted stable and a decrease of around 30% was observed after 35 days at room temperature. BSA and thrombin were tested as interfering no-target proteins and no significant electrochemical response was evidenced. Reproducibility and repeatability data were not provided and the aptasensor was not applied to real food samples.

Ensafi and co-workers developed two label-free electrochemical aptasensors for Lys detection, based on two different nanocomposites. In the first example, a GCE was modified with a nanocomposite including rGO, MWCNTs, CHI, and amino-functionalized carbon quantum dots (CQDs) synthesized from CHI [141]. The synergistic action of the different nanomaterials improved the electrical conductivity and electrocatalytic activity and sped up the electron transfer rate. In addition, it represented a proper sensing platform for the aptamer immobilization using covalent coupling between amino groups of the nanocomposite and those of the aptamer and glutaraldehyde as a linker. The Lys detection was followed by means of DPV and EIS. In the presence of Lys, when the immobilized aptamer selectively interacted with the target, a decrease in the DPV peak current and an increase in Rct in EIS were observed. Using the DPV and EIS data, two calibration curves were obtained with two linear concentrations ranges from 20 fmol L^−1^ to 10 nmol L^−1^, and 10 fmol L^−1^ to 100 nmol L^−1^ and two LODs of 3.7 and 1.9 fmol L^−1^ from DPV and EIS, respectively. The aptasensor’s reproducibility and repeatability were investigated with acceptable results in terms of RSD%: 4.2 and 3.8% (reproducibility, from DPV and EIS data, respectively) and 4.7% (repeatability). After one month at 4 °C in dry conditions, a signal response decrease of only 9% was recorded. BSA, human IgG, BHb (bovine hemoglobin), and thrombin were used as interfering no-target proteins. Using a mixture of these proteins with Lys, no considerable changes in the electrochemical signal response in comparison with that of Lys alone were evidenced. The aptasensor was applied to real spiked samples of egg white and wine, with satisfactory recoveries ranging from 94.0 to 96.2% (wine) and from 95.4 to 104.0% (egg).

In the second example, an SPCE was modified with a nanocomposite involving amino-reduced graphene oxide (amino-rGO), an ionic liquid (IL, 1-Butyl-3-methylimidazolium bromide), and amino-mesosilica nanoparticles (amino-MSNs) [142]. The nanocomposite morphological investigation revealed that the amino-MSNs were well distributed onto the rGO sheets. The synergistic action of the different nanomaterials improved the chemical and thermal stability, the conductivity, the electrocatalytic activity, and the biocompatibility of the electrodic material. The amino functionalities of rGO and of MSNs and the oxygen defects onto rGO supported the aptamer immobilization via covalent coupling, involving again glutaraldehyde as the linker.

The Lys detection was followed by means of DPV and EIS, as described in the previous example [141]. As already reported, when the aptamer selectively interacted with Lys, a decrease in the DPV peak current and an increase in Rct in EIS were observed. Using the DPV and EIS data, two calibration curves were obtained with two linear concentration ranges from 10 fmol L^−1^ to 200 nmol L^−1^ and 10 fmol L^−1^ to 50 nmol L^−1^ and two LODs of 2.1 and 4.2 fmol L^−1^ from EIS and DPV, respectively. The aptasensor selectivity was analyzed using a protein mixture of BSA, human IgG, prostate-specific antigen (PSA), BHb, and thrombin and no considerable changes in the electrochemical response in comparison with that of Lys alone were evidenced. The aptasensor reproducibility was investigated with satisfactory results in terms of RSD%: 4.85% (from DPV data) and 4.3% (from EIS data). The repeatability was considered obtaining acceptable results in terms of RSD% (5.1%). After one month at 4 °C in dry conditions, a signal response decrease of only 12% was recorded. The aptasensor was applied to real spiked samples of egg white and wine, with recoveries ranging from 94.6 to 96.0% (wine) and from 96.0 to 104.2% (egg). As a final comment, the two aptasensors showed similar analytical performances also in spiked real food products and no particular improvement was evidenced using different nanocomposites and electrode typologies, even if the linearity range seems to be improved in the last example.

Vasilescu and co-workers developed an aptasensor for Lys detection in wine, based on a AuSPE-modified with electrodeposited AuNPs, so increasing the conductivity and the electroactive surface of the electrode [143]. A thiol-modified aptamer was then chemisorbed on AuNPs and [Fe(CN)_6_]^4^^−/3^^−^ was used as the redox probe. The aptamer–lysozyme complex produced steric hindrances, limiting the diffusion and consequently the electron-transfer of the redox probe and a decrease in the [Fe(CN)_6_]^4^^−/3^^−^ electrochemical signal was recorded by CV. Such a decrease was considered as the analytical response and correlated to the Lys amount. An LOD of 0.32 μg mL^−1^ (23 nM) and a linear range between 1 and 10 μg mL^−1^ (70–700 nM) were obtained. After one month at 4°C and/or at −20 °C in dry conditions, no significant electrochemical response decreases were recorded. The reproducibility and repeatability were considered, obtaining acceptable results in terms of RSD%: 11.5% and 10.7%, respectively. Cytochrome C, BSA, neutravidin, and k-casein were tested as possible interfering proteins and no significant response was provided in red wines, while in white wine neutravidin gave a response due a non-specific adsorption, which can be eliminated subtracting it from the blank response. Finally, the aptasensor was applied to spiked real samples of wines with recoveries ranging from 82.3% to 92.3% and the results were comparable with those coming from HPLC.

Marrazza and co-workers reported a sandwich-format aptasensor including an SPCE modified with gold nanoclusters (AuNCs) and a conductive polymer such as poly-L-lysine [144].

We would like to highlight that the same group developed an aptasensor for β-LB detection using GSPE modified with poly-L-lysine and AuNPs [114]. The conductive polymer was electrodeposited onto the electrode surface and then AuNCs were deposited on it in the presence of 10,000 polyethylene glycol (PEG), acting as a stabilizer and dispersing agent of the nanoclusters and minimizing non-specific absorption. A schematic representation of the aptasensor is reported in Figure 12.

After the nanocomposite morphological and electrochemical characterization, a thiol-modified aptamer for Lys was immobilized by overnight incubation to form a self-assembled monolayer through the thiol–Au interactions. A sandwich assay was obtained after adding Lys and a secondary aptamer labeled with streptavidin alkaline phosphatase (S-AP). Under optimized experimental conditions, the Lys detection was carried out by means of DPV and a linearity range of 70–7 × 10 ^5^ pM with an LOD of 2 pM were achieved. Glucose and BSA were selected as possible interfering molecules because they are present in the same real matrices together with interleukin-6, having a similar molecular structure to that of Lys. No significant signal differences were observed in the presence of these interferences. Unfortunately, stability, selectivity, and repeatability data were not given.

The aptasensor was applied to spiked real samples of wines with satisfactory results in terms of recoveries, ranging from 97.4 to 109.7%. The results were comparable with those obtained from the commercial Qubit^®^ Fluorescence Protein Assay Kit (recoveries ranging from 97.5 to 102.8%).

Recently, a label-free origami microfluidic electrochemical nano-aptasensor was developed for the Ara h1 detection [145]. In Section 4.1, we already reported the characteristics and properties of the Ara h 1 allergen. The microfluidic aptasensor was realized involving the sequential folding of a piece of chromatographic paper substrate patterned with a microchannel, screen-printed electrodes, and pads for the loading sample. Black phosphorus nanosheets (BPNSs) were used because of their unique and reactive edge structure and high bandgap, improving and amplifying the electrochemical response. BPNS were functionalized with poly (lysine) to switch the negatively charged surface of BPNSs to positive to facilitate the aptamer immobilization through electrostatic interactions. Finally, the aptamer-decorated BPNSs were deposited onto the paper-based electrode surface as sensing probes. A scheme of the aptasensor assembly and its working principle is illustrated in Figure 13.

In the presence of Ara h 1, the target protein–aptamer complex steric hindrance limits the diffusion and the corresponding electron transfer of [Fe(CN)_6_]^4^^−/3^^−^ as the redox probe and a decrease in the redox probe peak current was recorded by DPV. Such a decrease was considered as the analytical response and correlated to the Ara h 1 amount. A linearity range of 50–1000 ng mL^−1^ with an LOD of 21.6 ng mL^−1^ was obtained. The repeatability was studied with acceptable results in terms of RSD% (4.2%). After one month at 4 °C, a signal response decrease of only 2% was evidenced. Some interfering proteins, including Ara h 2 and OVA were tested, but no significant electrochemical response was found. The microfluidic device was applied to real spiked cookie dough samples with a recovery range of 98.3–107.9% and relative standard deviations less than 5%.

As a general comment regarding the reported examples of aptasensors for allergen detection, we can observe that the LODs, independently of the analyte, achieved ng mL^−1^ or fg mL^−1^ in several examples. Concerning the format, generally label-free is preferred.

It must be underlined that almost all the aptasensors involved screen-printed electrodes and were applied to real samples and validated with reference analysis methods and in some cases validation was performed in certified external laboratories. This represents an important step forward if the purpose is to introduce the sensors in real life.

The analytical performances of the reported aptasensors for the determination of allergens as well as the corresponding sensor formats are summarized in Table 2.

### 4.3. Genosensors

Genosensors involve usually immobilized DNA/RNA probes as a recognition element and specific hybridization reactions related to the corresponding DNA–DNA or DNA–RNA molecular recognition. Examples of electrochemical genosensors are widely spread in the literature [146], in particular because of their potential for miniaturization, sensitivity, low LODs, including simple sample pretreatment. Consideration of the disadvantages, higher costs, and a certain instrumental complexity have to be evidenced [146]. 

A typical electrochemical genosensor contains an electrode, a capture probe, and a reporter probe. A capture probe recognizes and binds to the target and is usually immobilized onto the electrode surface. However, it can also be immobilized on nanomaterials or on other biomolecules which can be used to modify and/or improve the electrode surface. A reporter probe includes a molecule that produces an electrochemical signal in response to an electrochemical reaction. Both the capture probe and reporter probe present high specificity to the target DNA. Common molecules used as probes (capture and reporter) include single-stranded oligonucleotides, aptamers, peptides, and DNA-related proteins. In some sensors, the capture and reporter probes could be combined together to improve the sensing platform [147].

Covalent bonding and cross-linking involving strong and stable interactions are the most common immobilization approaches, while adsorption is not widely used, even if it is the simplest way to incorporate genetic material into the transducer surface.

Two genosensor formats are widely diffused for DNA hybridization detection: (a) label-free and (b) label-based formats [148].

The label-free approach is based on the changes in the redox properties of DNA electroactive bases. The main principle in label-free electrochemical DNA detection is based on the interaction of free guanine and adenine moieties of the DNA probe with its complementary thymine and cytosine bases of the DNA target during hybridization, producing a lower electrochemical response than that before the hybridization. This approach is simple and easy to perform, but, on the other hand, high background current for non-specific adsorption of the DNA target and high guanine oxidation potential are involved.

Concerning the label-based format, a label can be introduced both on capture and reporter probes [147,148,149]. If the capture probe is labeled, the analytical response is usually based on the proximity of the label to the electrode surface and, consequently, the electrochemical response can change in the presence of the analyte since the distance between the label and the electrode is different because of the target–probe interaction.

Otherwise, if the reporter is labeled, the target–capture probe interactions produce a sandwich-like structure, and then the corresponding electrochemical response is correlated to the presence and to the amount of label itself. A very interesting and accurate review concerning the state of the art of genosensors is available in the literature [149].

The Pingarron group developed an amperometric genosensor for the detection of Cor a 9, an allergen present in hazelnut [150]. Cor a 9 is an 11S globulin or legumin and is classified as a seed storage protein. Legumins such as vicilins and 2S albumins are particularly abundant in legumes and tree nuts and are thermostable and resistant to the digestion in the human body [90,94]. In addition, it is well known in the literature that Cor a 9 can be correlated with severe hazelnut allergy in children and in adults [151]. Coming back to the genosensor, the amperometric genosensor platform for hazelnut-trace detection involved the detection of PCR amplicons obtained from the hazelnut Cor a 9-allergen coding sequence by using an innovative Express PCR amplification. The proposed detecting system was based on a sandwich hybridization format for the detection of PCR products, carried out at the surface of streptavidin-modified MBs, using biotinylated capture and reporter probes. The magnetic bio-conjugates labeled with HRP were captured onto the SPCE surface.

Under optimized experimental conditions, a linearity range of 0.0024–0.75 nM and an LOD of 0.72 pM were achieved. The genosensor reproducibility including the complete sensor assembly and sensing protocol was evaluated with an acceptable result in terms of RSD% (6.2%). In addition, the long-term stability of the bio-conjugated MBs was analyzed and after 30 days at 4°C no significant changes in the genosensor electrochemical response were evidenced. Considering the coupling with Express PCR, denatured PCR amplified products were analyzed with the developed DNA sensor, using genomic DNA extracts from hazelnut. RSD% values were below 10%, confirming the good accuracy of extraction, amplification, and analytical protocols. The selectivity for hazelnut extracts was evidenced by carrying out the PCR amplification of DNA extracts coming from different hazelnut varieties and from other fruit and nuts and analyzing the resultant amplicons using the amperometric genosensor.

The next example considers the development of an amperometric genosensor for Sola l 7 allergen [152]. Sola l 7 belongs to the family of non-specific lipid transfer proteins (nsLTPs) and is present in tomato seeds [153,154,155].

Non-specific LTPs are involved in the plant growth process and in the defense mechanism against bacteria, fungi, and viruses, but this last aspect is not fully understood and clarified [153]. They are identified as allergens, attracting increasing interest nowadays. Particularly, in tomato seven nsLTPs are present and three of them are considered allergens [153,154,155]: Sola 1 3, Sola 1 6, and Sola 1 7. It is well known that they are stable in thermal treatments and resistant to the human digestion process, so they can continue to act as allergens in cooked and processed foods. Sola 1 7 was assumed as the most allergenic protein in tomato seeds. The Pingarron group developed a disposable electrochemical genosensor for PCR-free detection of Sola l 7, using DNA/RNA heterohybrids coming from a sandwich hybridization of a specific fragment of the Sola l 7 allergen coding sequence with appropriate RNA probes. Briefly, MBs modified with a specific RNA capture probe were used to selectively capture the target DNA, then hybridized with a specific RNA detector probe. The resulting sandwiched DNA/RNA heterohybrids were recognized by specific antibodies and then conjugated with secondary antibodies labeled with HRP. Finally, MBs with the sandwich RNA/DNA heterohybrids were magnetically captured on the SPCE surface, where the amperometric detection was carried out. In Figure 14, an illustration of the MBs-based amperometric biosensing strategy for targeting a specific fragment of the Sola l 7 allergen coding sequence is reported

After the optimization of all experimental parameters and conditions, a linearity range of 0.8–50 pM and an LOD of 0.2 pM were achieved. The reproducibility tests gave acceptable results in terms of RSD% (4.4%) and the long-term stability was investigated storing MBs modified with an RNA capture probe for 69 days at 4 °C and evaluating the amperometric responses of the corresponding genosensors every day; no significant changes in the electrochemical responses were evidenced. The selectivity was addressed considering the sensor response in the presence and in the absence of different DNA sequences such target DNA, single mismatched (SM), double mismatched (DM), triple mismatched (TM), and non-complementary (NC). Considering the responses vs. SM, DM, and TM sequences, a signal decrease of 23, 40, and 50%, respectively, was achieved, while no electrochemical signal was observed in the presence of NC; thus, indicating a good selectivity. Finally, the genosensor was applied to genomic DNA samples extracted from tomato peel and seeds and from corn. Corn contains Zea m 14, an allergenic nsLTP, with one of the highest percentages of similarity in its primary sequence with Sola l 7. Comparing the amperometric responses with the corresponding blanks responses, only those coming from the tomato seed genomic DNA were significantly different from the blank, underlining the selectivity and specificity of the genosensing approach.

As a final comment concerning the recent genosensors application for allergen detection, only two examples were included, but in my opinion they are very significant and interesting. The first one presented an integration between an electrochemical genosensor and PCR amplification evidencing the reliability of the analytical protocol including extraction, amplification, and amperometric detection and its applicability to the analysis of hazelnut extracts irrespective of variety. The more recent second example reported a sensing strategy using a sandwich hybridization format, producing quite long RNA/DNA heterohybrids and using a commercial antibody with a high affinity vs. RNA/DNA heterohybrids. This genosensing platform allowed tailoring the sensitivity by varying the bioassay format, heterohybrid length, or labeling strategy, without PCR or nanomaterial amplification. The analytical performances of the reported genosensor for the determination of allergens as well as the corresponding sensor formats are summarized in Table 3.

### 4.4. Cell-Based Biosensors

A cell-based electrochemical biosensor includes cells as sensitive recognition elements, able to react to an external stimulation or environmental changes [156,157,158]. Under the external stimulation, the cell response can be transformed in a measurable and processable electrochemical signal. Electrochemical cell biosensors can involve generally measurements of current, potential, impedance, conductivity, and capacitance. It is well known that cell-based biosensors can be used to study different cells, evaluating their typology and activity, but in this review, we reported electrochemical cell-based biosensors using immobilized living cells as the recognition element because under a specific stimulation they can modify their physiological state and this behavior is used for the detection of allergens in foods. The cell immobilization is a crucial step for the stability and the reliability of the biosensor. The most common method is to create a uniform adhesion layer on the electrodic surface, using an extracellular matrix (ESM) such as collagen, peptides, laminin, and SAM monolayer, among others. The traditional two-dimensional (2D) monolayer cell culture is the most diffused approach, producing an environment quite different from the natural 3D cellular environment; thus, also preventing cell-to-cell contact and covering the original morphological and functional characteristics of cells. Consequently, 3D cell culture has become an important methodological approach for assembling cell sensors, better simulating the in vivo environment for the cellular development. The role of ESM is fundamental for 3D cell culture, acting as a scaffold and frame.

Jiang et al. developed a disposable, electrochemical-mast-cell-based origami paper sensor for the detection of the milk allergen casein [159].

The group of milk proteins that precipitate include casein and it is identified as one of the major allergenic proteins in cow’s milk [160,161].

Coming back to the casein cell-based sensor, a nanocomposite including G, carbon nanofibers (CNFs), and gelatin methacryloyl (GelMA) was used to modify an SPCE; therefore, improving the conductivity and biocompatibility, and providing an appropriate sensing layer for the immobilization of rat basophilic leukemia (RBL-2H3) mast cells. It must be highlighted that paper can simulate the in vivo cell microenvironment as a 3D cell culture platform. This platform can control and monitor the interactions between cells and the cell responses to external stimulations. The casein antibody-sensitized mast cells were immobilized on the paper fibers through the biological affinity of the GelMA hydrogel. In Figure 15, a representation of the sensing platform assembly and assay procedure of casein is reported.

Summarizing the assembly strategy of the biosensor, the paper sheet was firstly patterned by means of a wax printer and then three electrodes were screenprinted on it. After modifying the electrodic surface with the nanocomposite and immobilizing the mast cell, the paper sheet was folded according to origami paper folding and integrated with the electrochemical device. The electrochemical investigation of the mast cells immobilized on the modified SPCE, using CV and DPV, showed an irreversible anodic peak and the corresponding peak current was proportional to the number of immobilized cells in the range from 1 × 10^2^ to 1 × 10^8^ cell /mL^−1^. After the biosensor interaction with casein, a significant decrease in the electrochemical response was observed, evidencing an inverse proportionality relationship between casein concentration and electrochemical response. A linearity range from 1 × 10^−7^ to 1 × 10^−6^ g mL^−1^ with an LOD of 3.2 × 10^−8^ g mL^−1^ was achieved. Considering the reproducibility, the RSD% values for the parallel detection of 1 × 10^−7^, 1 × 10^−6^, and 1 × 10^−5^ g mL^−1^ casein with 10 paper sensors, were 3.44%, 3.18%, and 3.51%, respectively. The long-time stability of the paper sensor (modified with only the nanocomposite but without mast cells) was studied after 21 days at room temperature and a decrease in the electrochemical response lower than 5% was obtained. BSA, OVA, soybean globulin, and shrimp TPM were tested as possible interfering proteins and they did not affect the casein detection.

The same Jiang group reported a paper-based capacitance mast cell sensor for real-time monitoring of the peanut allergen Ara h 2, a 2S albumin [90,91] like Cor a 14 (hazelnut) and Sin a 1 (mustard seed) [162]. A 3D paper chip printed with carbon electrodes was realized as a non-contact capacitance sensing platform and a composite hydrogel (PGHAP gel), containing polyvinyl alcohol (PVA), gelatin methacryloyl (GelMA), and nano-hydroxyapatite (nHAP), was employed to improve the conductivity and the biocompatibility of the paper. PGHAP gel and paper fibers acted as a 3D culture cell system and as an immobilizing platform for the RBL-2H3 mast cells. Briefly, the RBL cells were sensitized by an Ara h 2 antibody and immobilized on paper fibers modified with PGHAP gel and then the paper chip was incorporated in a polydimethylsiloxane (PDMS) holder for performing the capacitance measurements. The mast cell surface presented recognition receptors able to link selectively to the allergen protein, triggering the cellular degranulation and cellular content release. The alterations in the biochemical properties of these immobilized mast cells affected their capacitance, allowing real-time monitoring of allergens, which induced such sensitization effects.

Finally, the cellular degranulation and the cellular content release evaluation allowed the indirect quantitative detection of the allergen amount. Ara h 2 was determined in the concentration range of 0.1–1 ng mL^−1^ with an LOD of 0.028 ng mL^−1^. The capacitance sensor was applied to real samples of raw and fried peanuts and the results were comparable with those coming from the methods reported in the literature [162].

Finally, we would like to introduce a biomimetic “intestinal microvillus” electrochemical cell sensor for the detection of gliadin [163]. It was realized by means of 3D bioprinting, involving as ink ingredients, self-assembled flower-like copper oxide nanoparticles (FCONPs) and hydrazide-functionalized multi-walled carbon nanotubes (MWCNTs-CDH).

In particular, 3D bioprinting is a cross-science closely related to medicine, biology, mechanical engineering, and material science and can be assumed as a manipulation and treatment of bioinks to create or mimic living structures [164,165,166]. In the gliadin biosensor, the bioink was a conductive biocomposite hydrogel where FCONp and MWCNTs-CDH were incorporated in GelMA gel, which contains the arginine–glycine–aspartic acid (RGD) sequence suitable for cell adhesion. The biocomposite hydrogel was conductive for improving the biosensor analytical performance and presented a biocompatible 3D structure suitable for cell adhesion. The microvillus structure was printed on an SPCE with the bioink, and then the RBL-2H3 cells sensitized with a gliadin antibody were immobilized onto the microvillus. In Figure 16, a scheme of the assembly and detection procedures of the cellular electrochemical sensor is shown.

After the optimization of the immobilization conditions of the RBL cells, the analytical performances of the sensors were investigated by means of EIS. The EIS signal initially increased because of the electron-transfer hindrance between the gel and the electrode due to the RBL cell immobilization. In the presence of gliadin, with the increase in its concentration, the impedance signal decreased, indicating that as the target protein amount increased, a large number of cells died (apoptosis) and consequently were detached and fell off from the surface of the gel, restoring the electron transfer between the gel and electrode. A linear concentration range of 0.1–0.8 ng mL^−1^ and an LOD of 0.036 ng mL^−1^ were found. The reproducibility data were considered satisfactory with RSD% lower than 5%. The long-time stability of the sensor (without RBL cell immobilization) was studied for 24 days, with a decrease in electrochemical response of 2.6%, and after cell immobilization, a decrease of 2.7% was observed, always considering 24 days of storage. BSA, Ara h 1, OVA, soybean globulin, and shrimp TPM were tested as possible interfering proteins and they did not affect the gliadin detection. Corn flour, casein, rice flour, and Lys were tested for investigating the cell sensor specificity and the obtained results indicated a good sensor specificity. The cell-based sensor was applied to spiked real samples of gluten-free flour and gluten-free cookies, with recoveries ranging from 95.38% to 105% with RSD% of 3.28%.

As a final comment, the reported cell-based biosensor examples indicate an evolution of the allergen sensing approach towards a new generation of biomimetic electrochemical cell sensors. The 3D bioprinting technology can limit the drawbacks of artificial modified screen-printing electrodes, but above all 3D bioprinting can reproduce the structural features of native organs on a macro scale and provide a simulation of the allergic reaction in a more real environment, better reproducing the physiological state and processes of cells. 

The analytical performances of the reported cell-based biosensors for the determination of allergens as well as the corresponding sensor formats are summarized in Table 3.

### 4.5. Bacteriophage-Based Biosensors

Bacteriophage-based biosensors represent an unusual and valid alternative to conventional electrochemical biosensor for allergen detection. We would like to introduce some concepts and information about this novel approach. Bacteriophages (or phages shortly) are commonly defined as viruses able to infect and replicate within bacterial cells but not to invade other cells and they contain genetic information such as DNA or RNA encapsulated in a protein coat. Phages are nanoparticles with defined geometry and dimensions and are involved in different application fields ranging from ecology to diagnostics and also including the phage-display technology. The driving concept of the phage-display technology is the wide capability to modify the phage surfaces. In particular, phages can be modified genetically to display a foreign peptide on their surface.

Thanks to these modifications, phage particles acquire new properties such as being able to bind to desired target analytes or materials. Consequently, it is possible to create phages displaying on their surface peptides with high affinity to particular target molecules through a selection of several typologies among phages. This method can be adapted and tailored for the development of new bioreceptors and can find wide application in the electrochemical biosensing area. The procedure of selecting the appropriate peptide with desired affinity for the target is called biopanning [167] where the selector, i.e., the target molecule, is immobilized onto a solid support and then the mixture of phages is added and phages displaying a peptide able to bind to the selector are trapped. This procedure including target immobilization, phage-binding, washing, elution, and sequencing, can be repeated and at the end, the displayed peptides with the best affinity for the target are identified. Phages obtained by biopanning with the proper affinity vs. the target can be used as biorecognition probes in biosensors.

In comparison with other biological recognition elements such as antibodies or aptamers, phages are cheaper, very specific, and easy to modify and to handle even if under severe experimental conditions [167,168,169,170].

Bacteriophages are usually immobilized on the electrodic surface as the bioreceptors to detect the target analytes. The immobilized phage particles must preserve their binding affinity to their specific targets. In addition, a reproducible and repeatable surface modification is important for assuring biosensors’ high stability, reliability, and sensitivity. The most common methods for phage immobilization include physical adsorption, chemical functionalization including covalent bonding and interaction, such as biotin–avidin coupling.

Park and co-workers developed a phage-based electrochemical biosensor for detection of OM [171]. OM (Gal d 1) is classified as the most important allergen in hen eggs and is a highly glycosylated protein [172]. It is well known that OM together with OVA is used as a clarification agent in wine, promoting the elimination of tannins [88].

The M13 phage is selected as a bioreceptor and is classified as a filamentous phage. It is considered safe for humans and more resistant to the environmental and experimental conditions than antibodies, aptamers, and cells [171]. Using the biopanning protocol, the two clone phages with the displayed peptides with the best affinity for the target were selected.

The two phages were immobilized onto Au electrodes with two different covalent bonding procedures and the phage immobilized via the EC/NHS protocol showed the best binding affinity vs. OM.

A linearity range of 1.55–12.38 μg mL^−1^ and an LOD of 0.12 μg mL^−1^ were obtained by means of SWV. The reproducibility was acceptable in terms of RSD% (4.9%). The stability was also investigated and after 9 days at 4° no significant decrease in the electrochemical response was evidenced.

The phage sensor was applied to real spiked samples of egg white and wine and the recoveries and RSD% were evaluated with acceptable results: 97.5–108% (recovery range) and 7.6–9.6% (RSD% range) for real egg samples, 97.2–103.8% (recovery range) and 2.2–5.4% (RSD% range) for white wine samples.

Summarizing, the idea of using bacteriophages in the biosensing area resulted very promising, mainly because phage display techniques increase the number and possibilities of application fields thanks to the fact that phages are able to bind to several target analytes, such as small organic molecules and proteins such as allergens. Moreover, the presence of different functional groups on the phage surface can improve the phage particle immobilization on electrodes.

However, there are many challenges to face and overcome, concerning, for instance, the transfer of the phage-based sensors out of the laboratory and, consequently, the proper and accurate investigation of possible interferences present in a real matrix. Although phages are generally easy to produce at low cost on a large scale, as already mentioned, their purification is quite expensive and time-demanding and, therefore, their cost as recognition elements is still too high.

The analytical performances of the reported bacteriophage-based biosensors for the determination of OM as well as the corresponding sensor format are summarized in Table 3.

### 4.6. Molecularly Imprinted Polymer (MIP)-Based Sensors

MIPs can be easily synthesized and tailored for the selective detection of target analytes and are defined as “artificial antibodies” [173]. They represent an alternative to common bioreceptors, because of their selectivity, stability, low cost, and simple and ad hoc synthesis procedure for determination of different targets, in comparison to natural and most common bioreceptors such as antibodies and aptamers for instance. MIPs are synthesized through polymerization starting from the functional monomer in the presence of the analyte called the template molecule, so the template molecule was enveloped in the polymer structure. At the end of the polymerization process, the template molecule was extracted, and cavities with dimensions, shapes, and orientations corresponding to those of the template were produced. MIPs can recognize small target molecules but also, for example, proteins and viruses [15,173].

As a first example, we would like to introduce an MIP-based sensor for detection of BSA using DEIS as the analytical technique and a GC electrode modified with CHI and PPY, acting as an MIP [174]. The DEIS technique was already described in Section 3.1 in terms of a combination of EIS and CV. CHI was casted on the electrode and MIP was synthesized employing pyrrole as the functional monomer and electropolymerization as the synthetic approach in the presence of BSA. Finally, BSA was removed just after the electropolymerization ended.

BSA is an allergenic protein present in bovine blood plasma, as well as in beef and cow’s milk and it is also used as a food additive because of its emulsifying properties [175,176,177]. Under optimized experimental conditions, the linearity range of 0.0001–1 ng mL^−1^ and an LOD of 5 × 10^−5^ ng mL^−1^ were achieved. Human serum albumin (HSA) and BHb were tested as possible interfering proteins because they were chemically and structurally similar to BSA. HSA and BHb evidenced a lower binding effectiveness than that of BSA, indicating a good sensor selectivity.

The reproducibility was considered satisfactory with an RSD% of 2.1%. Concerning the operational stability, the sensor was used for six consecutive analyses with acceptable results in terms of RSD% (1.7%). Concerning the long-term stability, after 10 days at 4°C in the refrigerator, a decrease in the electrochemical response of only 1.75% was found. The MIP sensor was applied to spiked real samples of human blood serum with a recovery range from 98 to 102% and with RSD% ranging from 0.9 to 2.6%. Finally, these data resulted comparable with those coming from HPLC.

An electrochemical sensor based on MIP was developed for β-LB detection using choline chloride as a functional monomer, β-LB as the template molecule, ethylene glycol dimethacrylate (EDMA) as the cross-linking agent, and benzoyl-NN-dimethylaniline (BPO-DMA) as the polymerization initiator [178]. Next, the polymer was immobilized onto an SPCE modified with a nanocomposite including polyethyleneimine (PEI)-reduced graphene oxide (rGO) and AuNCs, as illustrated in Figure 17.

The synergistic action of the different nanomaterials enhanced the electrocatalytic activity and improved the electron transfer rate. After the optimization of the experimental conditions, β-LB was determined by means of DPV obtaining a linear concentration range of 10^−9^–10^−4^ mg mL^−1^ and an LOD of 0.02 mg mL^−1^. OVA, BSA, casein, and thermally denatured β-LB were considered as interfering molecules and no significant change in the electrochemical response was observed after the addition of interferences, revealing the sensor specificity. The reproducibility was investigated with acceptable results in terms of RSD% (2.2%). The sensor was applied to spiked real samples of milk and the obtained data was comparable with those coming from the ELISA conventional method.

An MIP-based electrochemical sensor for Cor a 14 detection was assembled using a AuSPE modified with electropolymerized PPY as the MIP and Cor a 14 as the template molecule [179], as illustrated in Figure 18. 

After optimizing the electropolymerization conditions and parameters and the template molecule amount, a linearity range from 100.0 fg mL^−1^ to 1.0 mg mL^−1^ and an LOD of 24.5 fg mL^−1^ were obtained, using SWV as the electroanalytical technique. Cor a 14 is classified as 2S albumin; thus, different 2S albumins from other plant sources, such as tree nuts, as well as from legumes and cereals, were selected as possible interferents because I do not see where I have to change the hyphen of their similarities in the structure to Cor a 14. The selectivity of the MIP sensor was also evaluated vs. proteins of animal origin, such as milk proteins including casein and the whey proteins. The MIP sensor had good specificity, but must be underlined that the specificity and selectivity were lower if the 2S albumins were genetically related to Cor a 14 [179.]. In addition, a comparison of the MIP selectivity with the corresponding immunosensor developed by the same research group using anti-Cor a Cor14 IgG (raised in rabbit) [91], as already described in Section 4.1, is reported. The selectivity performance of the two sensors was equivalent, so Cor a 14-MIP can be considered effectively as an antibody-like recognition element. The MIP sensor was applied to mixtures of pasta containing known amounts of hazelnut as models of real samples. The electrochemical MIP sensor determined 0.16 mg kg^−1^ of Cor a 14 in 1.0 mg kg^−1^ of hazelnut in pasta, which was the same LOD reported for the corresponding electrochemical immunosensor in Section 4.1 [92]

The next example described an MIP sensor for the qualitative detection of genistein as a marker for the presence of soy allergenic proteins; thus, overcoming the issues related to the detection of proteins including high cost, stability, fermentation, and so on [180]. It is well known that soy is an allergenic food. In fact, several soy proteins such soy hydrophobic protein, soy hull protein, soy profilin, and soy glycinin can induce adverse reactions and consequently are responsible for soy allergy as reported in Section 4.1. On the other hand, soy also contains several isoflavones and among them, genistein is one of the isoflavones present in a higher amount. Moreover, the genistein amount present in soy food products is independent from the processing and preparation methods [180], so it can be considered as a soy marker. Briefly, the MIP sensor assembly involved the electropolymerization of *o*-phenylenediamine (*o*-PD) on an SPCE, in the presence of genistein. The template molecule was removed just after the electropolymerization ended. Genistein was detected by means of DPV in the concentration range from 100 to 10 ppm with an LOD of 100 ppb.

The long-term stability was investigated and after 10 days at 65 °C a negligible decrease in the electrochemical response was evidenced. The relatively high temperature accelerated the MIP ageing process, reducing the long-term stability test time [180]. 7-hydroxyflavone, quercetin, and daidzein, as flavones and isoflavones, chemically and structurally similar to genistein, together with vitamin C and amino acids present in food products such as tryptophan and tyrosine, were tested as possible interfering molecules. The results indicated that it is possible to easily distinguish both among genistein and other molecular interferents present in food samples and among genistein and flavonoid antioxidants, structurally similar to it. The MIP sensor was applied to real food samples including store-bought products, home-baked goods, and restaurant dishes and the corresponding results were compared with those coming from the LF assay. In each case, the MIP sensors identified qualitatively the presence or absence of soy, in accordance with the LF assay results.

As the last example, an MIP-based sensor for the detection of gluten was introduced, including a combination of MIP with superparamagnetic iron oxide nanoparticles (SPIONs) [181]. SPIONs were synthesized via the chemical co-precipitation method and incorporated in the MIP structure during the chemical polymerization of methyl methacrylate (MMA) as a functional monomer in the presence of gluten as the template. After the template removal, a magnetic MIP (MMIP) was obtained and used to modify a carbon-plate electrode. 

SPIONs role was to improve the electron transfer to and from the electrode surface and, consequently, the sensor sensitivity. A scheme of the MMIP sensor assembly and of the corresponding gluten detection strategy is reported in Figure 19.

The gluten concentration was measured by means of amperometry in the range from 50 to 1000 ppm, with an LOD of 1.50 ppm, and as the gluten concentration increased, the corresponding electrochemical signal decreased because the gluten molecules can prevent the electron transfer. The reproducibility was investigated, and six electrodes showed comparable results in terms of LOD and sensitivity.

Concerning the operational stability, the MMIP sensor was tested and after six consecutive analyses at three different gluten concentrations, similar results in terms of amperometric current were achieved. Glutamic acid and glycine were employed as interfering amino acids and the interferents’ responses could be considered not significant with respect to that of gluten. The MMIP sensor was finally applied to a real sample of crackers, such as gluten-free original, gluten-free salt and vinegar, gluten-free seaweed, gluten-free barbeque, original, and wheat, evidencing qualitatively the presence or the absence of gluten.

Summarizing, it must be underlined that MIP represents a valid alternative to the most common bioreceptors and being artificial, the problems correlated to the stability of the bioreceptor are overcome. Finally, the analytical performances, including stability and selectivity, are comparable with those observed with more conventional biosensors for allergens such as immunosensors and aptasensors.

The analytical performances of the reported MIP-based biosensor for the determination of allergens are summarized in Table 3.

## 5. Conclusions

In this section, we would like to draw some conclusions regarding the biosensors, the electrode typologies, and the nanomaterial roles. In addition, some comments concerning detectability, detection limits, selectivity, and the validation of sensors with a standard method of analysis, and the possibility of determining several analytes at the same time, are outlined. Finally, critical issues, challenges, and future perspectives on the electrochemical biosensing approach for allergen detection are introduced and evidenced.

Firstly, some comments on the different biosensors are required. In this review, several examples of biosensors for allergen detection are reported, but most of them are immunosensors and aptasensors. The number of examples of immunosensors is comparable to that of aptasensors. It must be underlined that the analytical biosensing performances can be considered similar for aptasensors and immunosensors [182]. The aptamers are evaluated as having enhanced stability, involving a more flexible design, and being more suitable to be regenerated in reusable biosensors. For example, the regeneration step for immunosensors is complex and difficult and in any case, even if the target can be released from the immunosensor, the antibodies are irreversibly and irreparably damaged. On the other hand, it should be mentioned that aptamers, being nucleotides, are limited in their functional multiplicity with respect to the other biomolecules, and moreover the aptamer and antibody performances are strictly dependent on the immobilization protocol and orientation and from the transducer properties. As a final comment, antibodies are still considered as the first choice for a capture probe in biosensors, but the competition with the aptamers is increasing because the aptamers can overcome the immunosensors’ shortcomings, in terms of size, stability, variability between batches, cost, and flexible design, and they seem also to provide new ideas for enhancing biosensor analytical performance.

Considering the genosensor typology, very few example are present in this review, probably because sample preparation and reagent handling are still the main obstacle to overcome towards an up-to-date allergen detection, in addition requiring skilled personnel and sophisticated laboratory instrumentations. An innovative and interesting approach is reported in this review [152], where a genosensor that is easy to handle, low cost, and PCR-free can represent a step towards the routine and on-site detection of allergens.

Cell-based biosensors and bacteriophage-based biosensors can become promising alternatives in the electrochemical biosensing approach for allergen determination. In particular, the cell-based biosensors can be considered an evolution of the allergen sensing approach towards a new generation of biomimetic electrochemical cell sensors where 3D bioprinting technology can limit the drawbacks of artificially modified screen-printing electrodes [163]. Some critical issues remain to be solved regarding the specificity and stability of the sensor and the costs of cell culture; in fact, all these problems make it difficult to market cell-based biosensors.

Recently, phage-based electrochemical sensors have attracted attention because the development of the phage display technique allows to increase the use of these kinds of sensors. In fact, different genetically engineered peptides or proteins can be displayed on the phage surface; thus, binding many other possible target analytes. Unfortunately, although the cost of the phages and reagents are lower than those of other bioreceptors, e.g., antibodies, their purification is still expensive and time-demanding, and so it is necessary to reduce the phage purification costs for a phage-based sensor’s possible commercialization.

Finally, the MIP electrochemical sensors examples are limited, probably because a more traditional approach including a more conventional bioreceptor is preferred, even if the analytical performance with an “artificial antibody” based sensor resulted comparable with those obtained with BREs.

Most of the described biosensors use screen-printed electrodes. Electrochemistry and the corresponding analytical techniques combined with SPEs are capable of providing cost-effective, accurate, sensitive, and fast analytical tools for on-site analysis. In addition, the nanomaterials’ introduction together with the progress of biotechnology can be used to implement the SPE-based biosensing platforms for in situ monitoring. Finally, a brief comment about the environmental impact of screen-printed electrodes is required because they are disposable and single-use electrodes, involving a relatively high number of analyses. The environmental impact of substrate materials and of the electrode materials must be analyzed. As already reported in the literature [183], paper, glass, and ceramics are the best options to reduce the SPEs footprint as substrate materials, while, considering the electrode materials, the substitution of noble metals with carbon-based materials reduced the corresponding environmental impact. It must be underlined that most of the biosensors mentioned in this review employed carbon-based SPEs, involving graphite, CNTs, or graphene as the carbon material.

Considering several examples reported, the integration of nanomaterials in the design of electrochemical biosensors resulted as an added value for improving their analytical performance. Nanocomposites and/or nanohybrids represented the best option, including nanoparticles, nanotubes, nanofibers, or polymers, both natural and synthetic, and the corresponding nanostructures were very complex with tailored architecture. The combination of different materials, integrating the electron transfer capability, the conductivity, and the electrocatalytic properties enhance the analytical performance of the sensors.

The analytical performance of the biosensors, the linearity ranges, and the LODs achieved, generally the ng mL^−1^ and sometimes fg mL^−1^, regardless of the type of biosensor, its format, and the analyte were evaluated.

Sensor selectivity was addressed, but the choice criterion among potentially interfering compounds and/or macromolecules is not always clear; for instance, if interfering proteins should be selected because they have similar amino acid sequences and structures or belong to the same class of proteins with respect to the target analyte and/or are present in the same complex matrix to be analyzed. It would be important to indicate this criterion in order to compare selectivity data for the same target analyte, coming from different biosensors.

The biosensors’ reproducibility, repeatability, and stability were not always investigated, and the corresponding data are not comparable, even if considering the same target.

For example, analyzing the long-term stability data, the durability and the storage conditions are different from case to case and a proper comparison is not possible.

Generally, the biosensors reported in this review were applied to spiked real samples and it is a crucial step for introducing them in real life, but a validation with a conventional method such as ELISA or HPLC is mandatory for a clear and objective evaluation of the biosensors’ analytical performances.

As a general comment, the major challenge is to apply the allergen electrochemical biosensors in a real-world sample. The smart and reliable monitoring of allergens is required for food safety and this purpose can be achieved if allergens kits will be available for food producers and consumers, but unfortunately electrochemical biosensors for accurate and sensitive detection of allergens are not commercially provided. Several drawbacks can be considered starting from the stability of the biorecognition element and the cost of the materials used and the corresponding sustainability for the appropriate testing in a real and complex matrix. The implementation of the on-site analysis and the accurate evaluation of the real matrix can partially solve these issues, together with the miniaturization and the intelligentization of the biosensing devices and an appropriate integration of nanomaterials or nanocomposites.

Moreover, sample preparation represents a criticality when a solid sample is involved because the extraction procedures cannot be easily standardized, regarding the different physical chemical properties and the different stability of target analytes, and the different real matrices. 

A biosensing device for detecting of different allergens in the same sample or the same allergen in different samples can represent an effective analytical tool able to reduce the costs and the time taken for analysis.

In this review, the iEAT system represents an innovative biosensing system for a simultaneous detection of different allergens in a sample. In fact, it enables a fast, accurate, and cost-effective quantitative detection of five allergens in real food products, starting from packaged food and desserts to restaurants dishes [108]. Considering the consumer-friendly aspect, the extraction kit is simple to use, and the integrated communication protocols allow users to record and upload data in a cloud server. This electrochemical system indicated a development level very close to a possible market introduction and can be assumed as an ASSURED device (affordable, sensitive, specific, user-friendly, rapid, and robust and not-large electricity-dependent) according to the WHO guidelines [107].

The MIP sensor for the detection of genistein as a soy marker [180] can be considered both as an example of a sensing device for the analysis of a single allergen in different food samples and as an example where the allergenic protein is substituted as a target analyte with a more stable small organic molecule such as genistein. In addition, the issues due to the stability of the bioreceptor are overcome by using MIP as the “artificial antibody”. The MIP sensor enables the detection of genistein in real food samples such as restaurant meals, store-purchased food products, and home-baked goods, and the results were validated by means of LF assay. Starting from this approach, a so-called Allergy Amulet electrochemical sensing platform was developed [184], based on the same MIP sensor previously assembled for genistein detection. Allergy Amulet was applied to 42 food products such as grocery foods and restaurant dishes, containing more than 300 ingredients and correctly indicated the presence or absence of a soy allergen marker as confirmed by the LF kit. This sensing platform can represent another innovative tool towards an easy and user-friendly protocol for allergy detection and became a commercial kit in autumn 2021.

As a final comment, we can consider these last two sensors very promising and for different reasons they seem to approach marketing, but the road ahead is still long and difficult.

## Figures and Tables

**Figure 1 biosensors-12-00503-f001:**
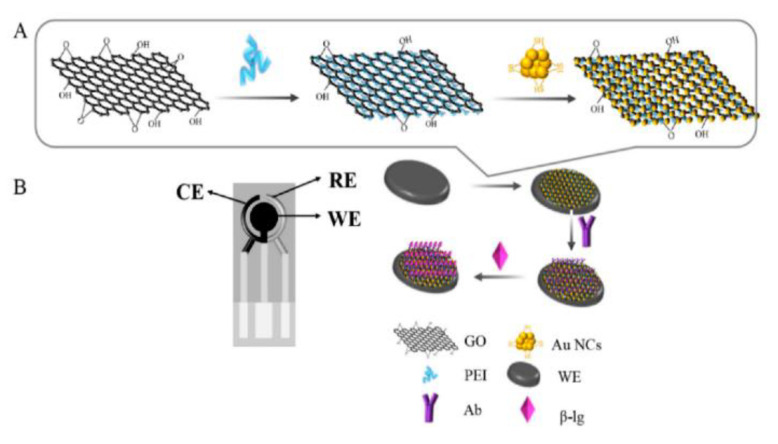
Immunosensor assembly strategy: (**A**) nanocomposite synthesis, modifying rGO with PEI and AuNCs (PEI-rGO-AuNCs); (**B**) SPCE modification, Ab immobilization, and β-LB detection, reprinted from [72]. CE = counter electrode; RE = counter electrode; and WE = working electrode.

**Figure 2 biosensors-12-00503-f002:**
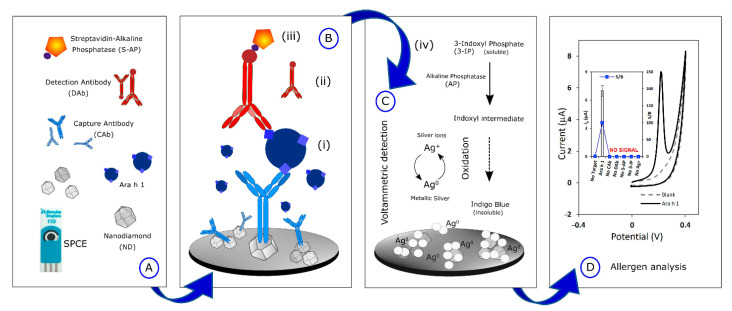
Scheme of the sensor assembly (**A**), the sensing mechanism (**B**), and the allergen detection (**C**,**D**) Allergen electrochemical detection. Reprinted from [95].

**Figure 3 biosensors-12-00503-f003:**
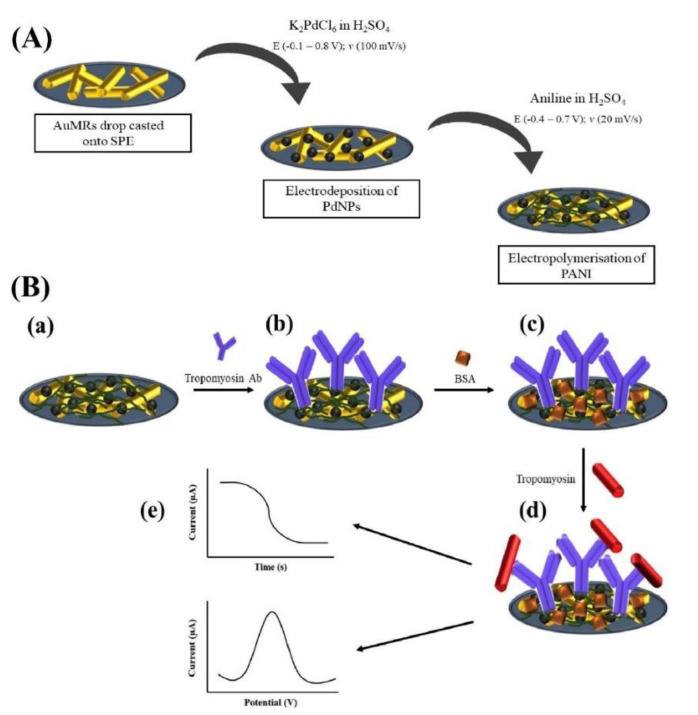
Scheme of the preparation of the SPCE modified with AuMRs/PdNPs/PANI composite and assembly of the TPM immunosensor: (**A**) AuMRs casted onto the working electrode, electrodeposition of PdNPs, and electropolymerization of PANI; (**B**) (**a**) AuMRs/PdNPs/PANI-modified SPCE; (**b**) TPM-Ab functionalized and immobilized onto the modified SPCE after EDC/NHS activation; (**c**) blocking of biosensor surface by BSA; (**d**) addition of different concentrations of TPM; (**e**) electrochemical measurements using CA and DPV techniques. Reprinted with permission from [98]. Copyright 2020, Elsevier.

**Figure 4 biosensors-12-00503-f004:**
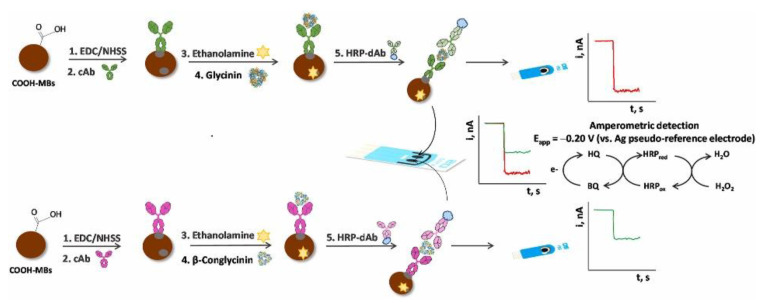
Schematic representation of the bioplatform assembly and the immunosensing mechanism involving the amperometric determination of β-conglycinin and glycinin at SPCE. Reprinted from [106].

**Figure 5 biosensors-12-00503-f005:**
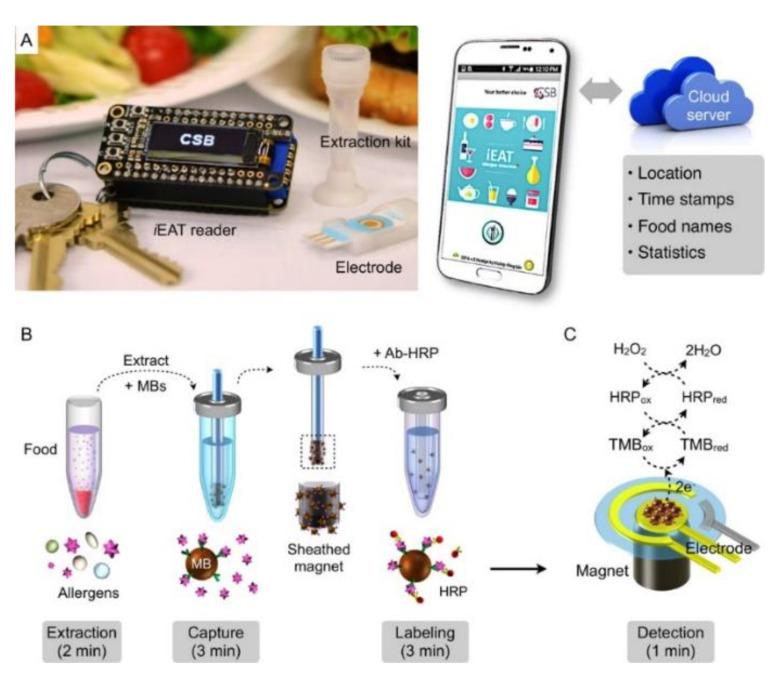
iEAT system for on-site allergen detection. (**A**) The system consists of a pocketsize detector, an electrode chip, and a disposable kit for allergen extraction. The detector is connected to a smartphone for system control and data upload to a cloud server. (**B**) Antigen extraction. Antigens are captured on MBs and then labeled with antibodies conjugated with HRP. (**C**) For signal detection, HRP-coated MBs are mixed with electron mediators (TMB, 3,3′,5,5′-tetramethylbenzidine) and dropped on the electrode. HPR catalyzes the oxidation of TMB. Reprinted with permission from [108]. Copyright 2017 American Chemical Society.

**Figure 6 biosensors-12-00503-f006:**
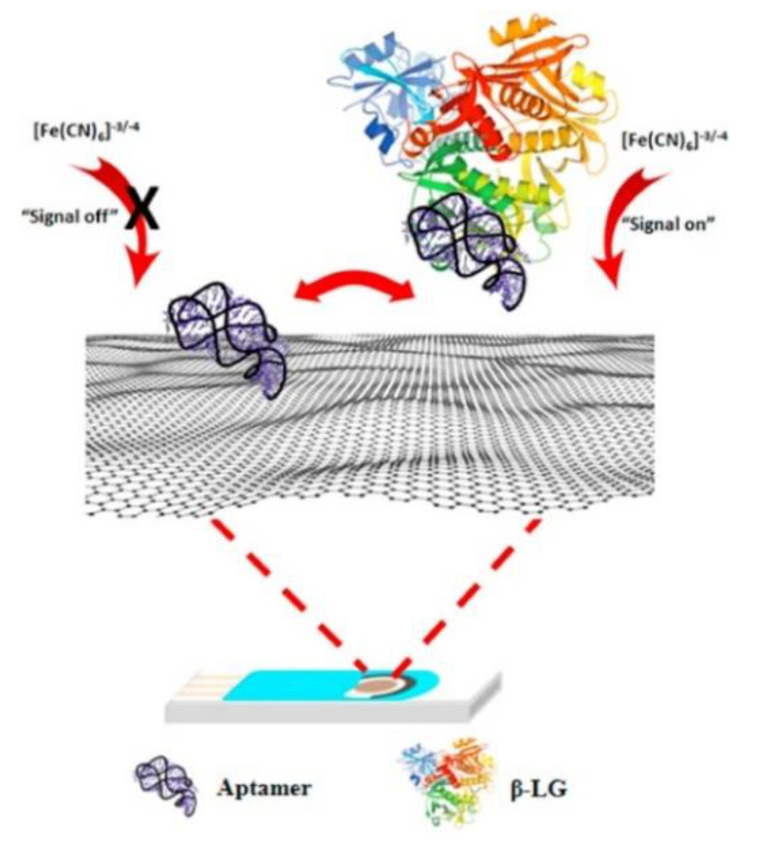
β-LB detection scheme based on aptamer-functionalized graphene screen-printed electrode. Reprinted with permission from [111]. Copyright 2017, Elsevier.

**Figure 7 biosensors-12-00503-f007:**
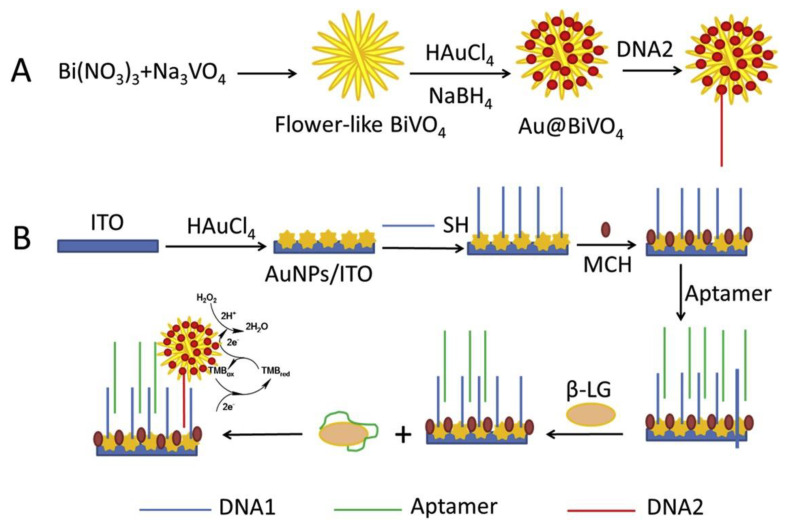
Scheme of the β-LB detection strategy. (**A**) Preparation of the DNA2/Au@BiVO_4_ probe; (**B**) assembly of the aptasensor and electrochemical detection of β-LB. Reprinted with permission from [117]. Copyright 2020, Elsevier.

**Figure 8 biosensors-12-00503-f008:**
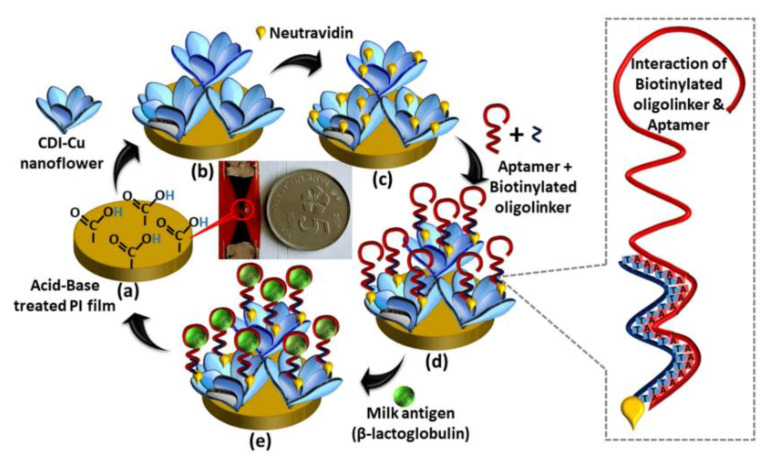
Stepwise realization of the β-LB aptasensor including polyimide (PI) film as the electrode substrate for LSGE, (**a**) acid–base treatment, (**b**) deposition of CDI-Cu hybrid NF, (**c**) modification with neutravidin, (**d**) immobilization of the aptamer, and (**e**) detection of β-lactoglobulin. Reprinted from [121].

**Figure 9 biosensors-12-00503-f009:**
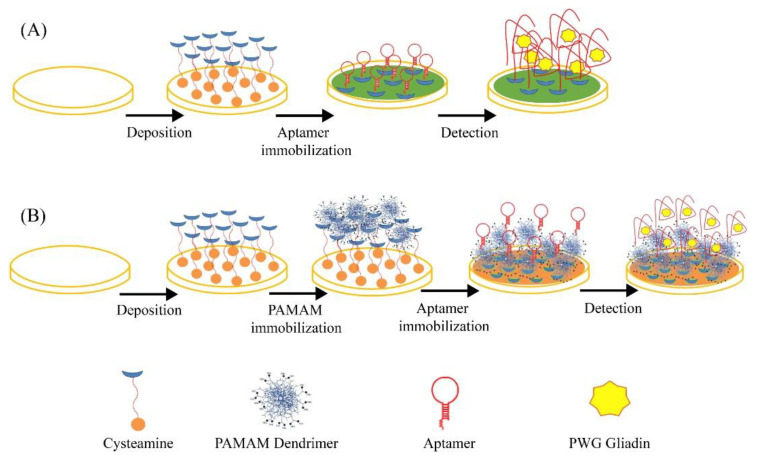
Scheme of the aptasensor (**A**) without PAMAM G4 and (**B**) with PAMAM G4. Reprinted with permission from [126]. Copyright 2017, Elsevier.

**Figure 10 biosensors-12-00503-f010:**
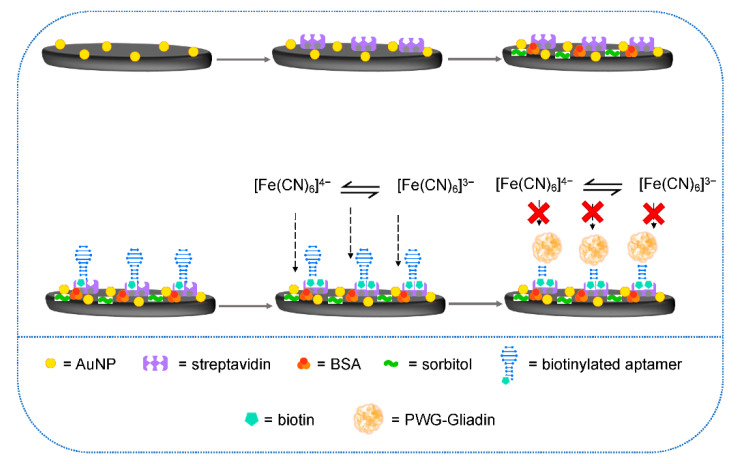
Scheme of the aptasensor design and sensing strategy: after the electrochemical deposition of AuNPs streptavidin was immobilized on the working electrode, then a solution of BSA and sorbitol was used to block the surface; subsequently, the biotinylated aptamer was immobilized, and after a final step of blocking with biotin, the sensor was ready to use. Reprinted from [130].

**Figure 11 biosensors-12-00503-f011:**
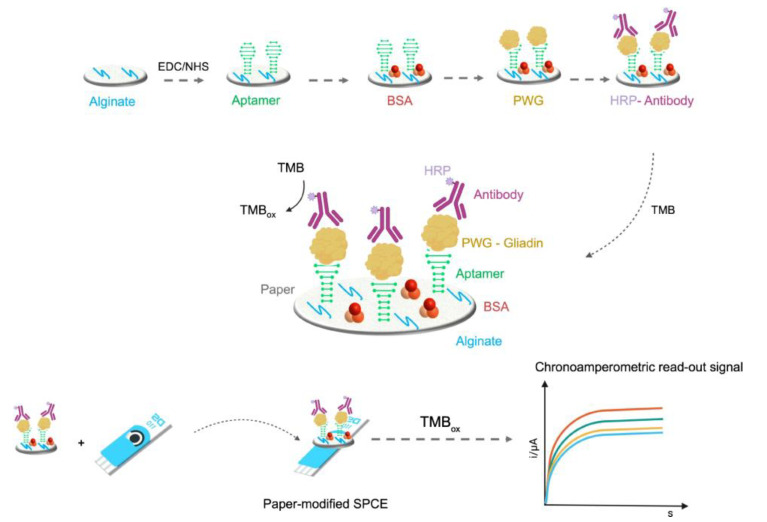
Schematic representation of the gluten paper-based biosensor design and working principle. Reprinted from [137].

**Figure 12 biosensors-12-00503-f012:**
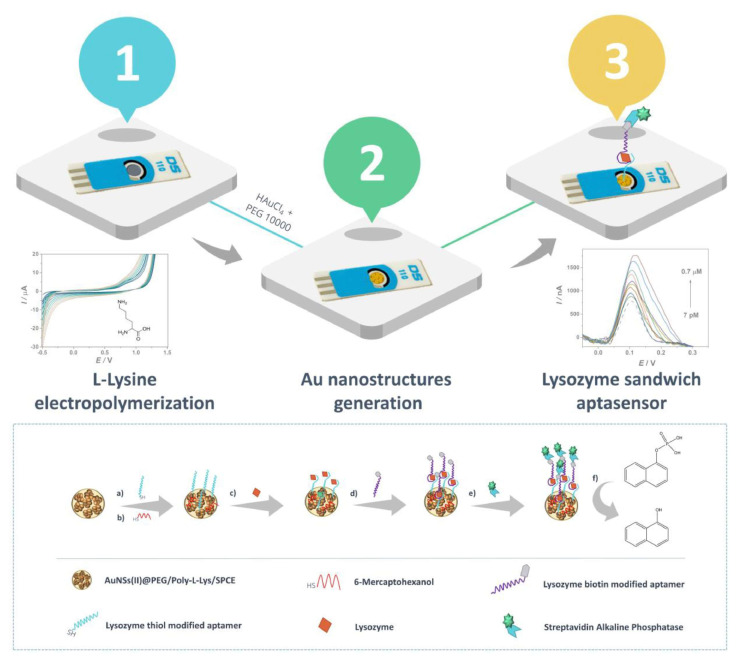
Scheme of the aptasensor assembly and the corresponding sensing mechanism. Reprinted with permission from [144]. Copyright 2022, Elsevier.

**Figure 13 biosensors-12-00503-f013:**
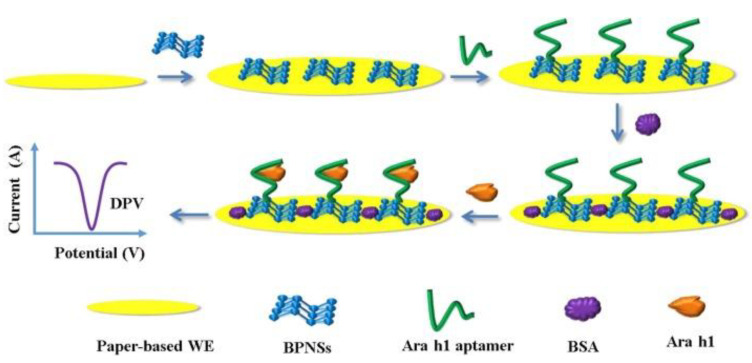
A schematic illustration of the aptamer sensing mechanism of the microfluidic origami electrochemical aptasensor. Reprinted with permission from [145]. Copyright 2021, Elsevier.

**Figure 14 biosensors-12-00503-f014:**
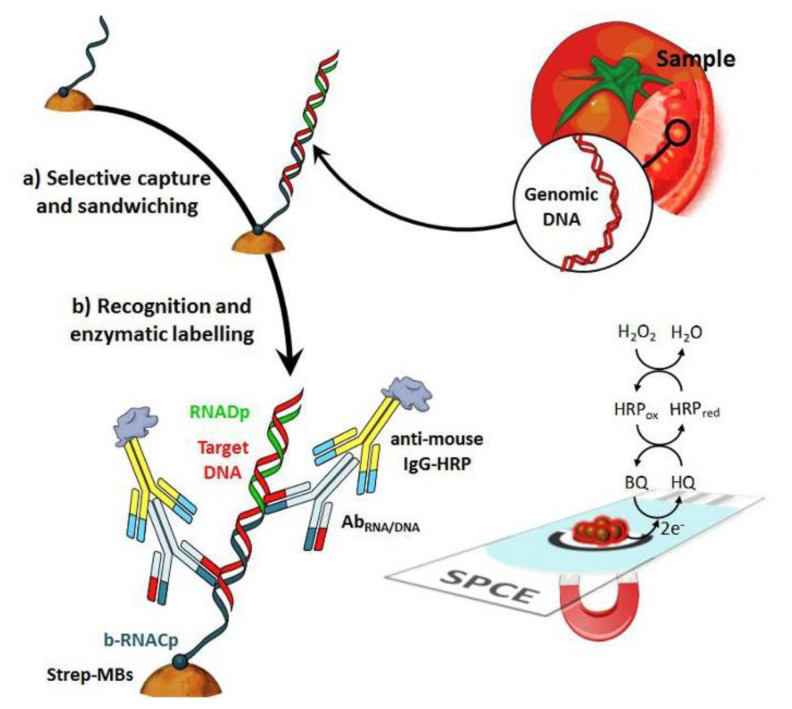
Scheme of the MBs-based amperometric biosensing strategy developed for detecting Sola l 7 allergen. Reprinted with permission from [152]. Copyright 2019, Elsevier.

**Figure 15 biosensors-12-00503-f015:**
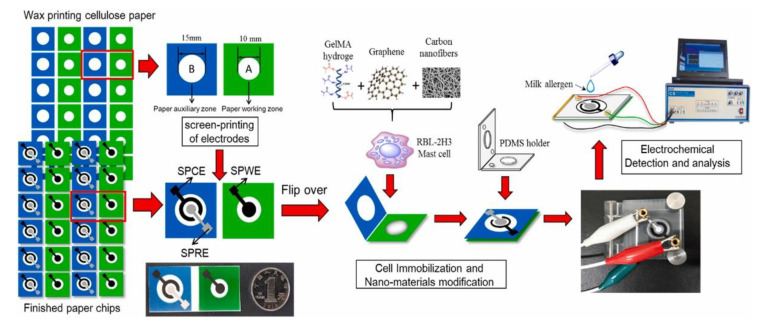
Scheme of the sensing platform assembly and assay procedure of casein detection. Reprinted with permission from [159]. Copyright 2019, Elsevier.

**Figure 16 biosensors-12-00503-f016:**
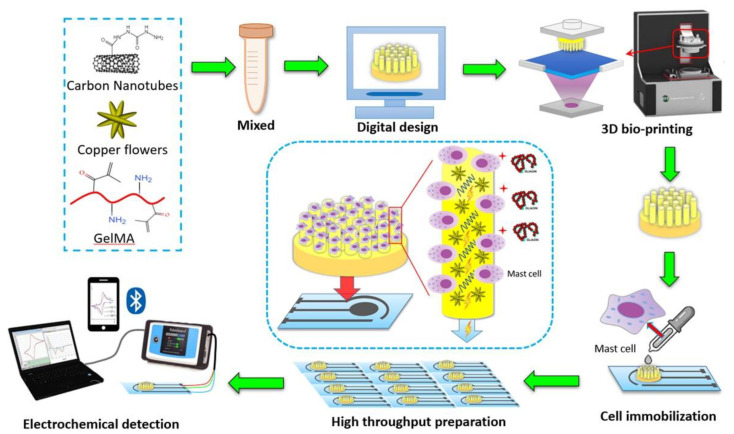
Schematic diagram of the assembly and detection procedures of the cell-based electrochemical sensor for gliadin detection. Reprinted with permission from [163]. Copyright 2021, Elsevier.

**Figure 17 biosensors-12-00503-f017:**
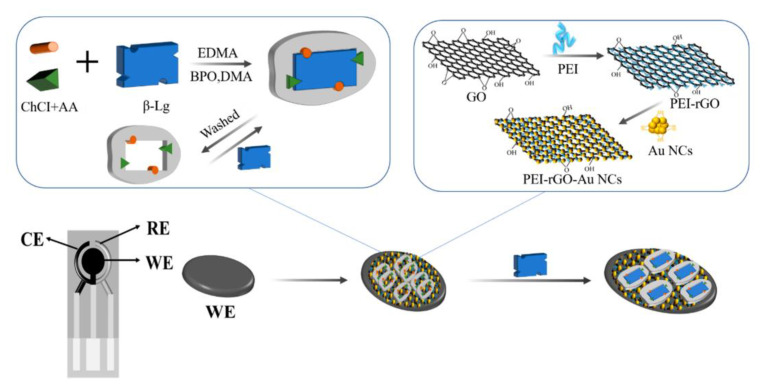
Scheme of the MIP synthesis and of the electrochemical sensor assembly. Reprinted from [178].

**Figure 18 biosensors-12-00503-f018:**
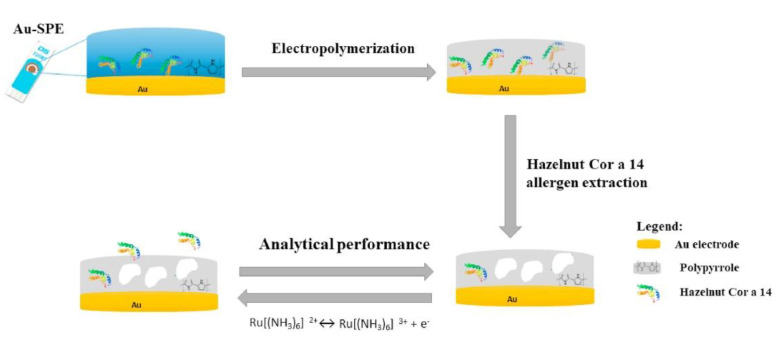
Scheme illustrating the different steps involved in the MIP assembly and electrochemical detection of hazelnut Cor a 14 allergen. Reprinted with permission from [179]. Copyright 2022, Elsevier.

**Figure 19 biosensors-12-00503-f019:**
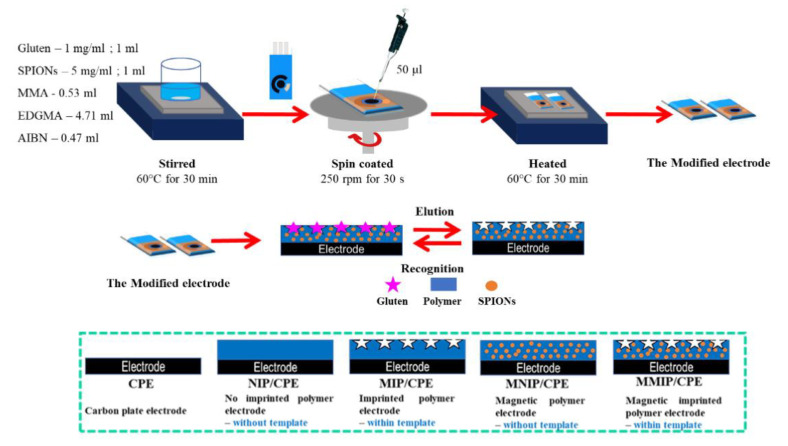
Scheme of the MMIP sensor assembly and the corresponding gluten detection strategy. Reprinted from [181].

**Table 1 biosensors-12-00503-t001:** Performance of electrochemical immunosensors for allergen detection.

Electrode	Immunosensor Format	Electrochemical Technique	Analyte/Sample	Linearity Range	LOD	Recovery (%)	Reference Method	Ref.
AuSPE	Label-free format based on Ab entrapment in PPY film	DPV	α-LB/milk	355–2840 pg mL^−1^	0.18 fg mL^−1^	93–97	-	[71]
SPCE	Label-free format based on Ab immobilization on PEI-rGO-AuNCs nanocomposite	DPV	β-LB/milk	0.01–100 ng mL^−1^	0.08 ng mL^−1^	-	ELISA	[72]
SPCNTE	Sandwich format using an immobilized primary Ab on SPCNTE and an HRP-labeled secondary Ab	Amperometry	β-LB/-	Sub ppm–10 ppm	0.173 ppm	-	-	[73]
GCE	Label-free format based on Ab immobilization in collagen film	EIS	Gliadin/-	5–20 mg L^-1^	5 mg L^−1^	-	-	[74]
SPCE	Sandwich format using a primary Ab immobilized on a paper platform located on CNFs/SPCE and an HRP-labeled secondary Ab	Amperometry	Gliadin/flours	0–80 μg kg^−1^	0.005 μg kg^−1^	98.5–102.10	ELISA	[76]
Ti	Label-free format including Ab immobilization on TiO_2_NTs-GO nanocomposite	EIS	Gliadin/-	0–20 ppm	14 ppm	-	-	[78]
SPCE	Sandwich format involving primary Ab immobilized on MBs and HRP-labeled secondary Ab	Amperometry	OM/egg white, wheat flour, bread	0.3–25 ng mL^−1^	0.1 ng mL^−1^		ELISA	[82]
SPGE	Label-free format including Ab immobilized on Fe_3_O_4_@PdNPs/CHI nanocomposite	DPV	OVA/food samples	0.01 pg mL^−1^–1 μg mL^−1^	0.01 pg mL^−1^	101.6–107.0	-	[87]
SPCE	Sandwich format involving primary Ab immobilized on GO/PDDA-modified SPCEs and the HRP-labeled secondary AB immobilized on MBs	Amperometry	OVA/wine	0.01–10 pg mL^−1^	0.2 fg mL^−1^	-	ELISA	[89]
AuSPE	Label-free format including Ab immobilized on SAM modified electrode surface	SWV	Cor a 14/wheat flour	0.1 fg mL^−1^–0.01 ng mL^−1^	0.05 fg mL^−1^	-	-	[92]
SPCE	Sandwich format involving capture Ab immobilized on NDs-modified SPCEs and S-AP-labeled secondary Ab	ASV	Ara h 1/Biscuits, crackers, cookies, cereals, energetic protein bars	25–500 ng mL^−^^1^	0.78 ng mL^−^^1^	-	ELISA	[95]
SPCE	Sandwich format involving primary Ab immobilized on MNPs and HRP-labeled secondary Ab	Amperometry	TPM/-	0–218.7 ng mL^−1^	46.9 pg mL^−1^	-	-	[97]
SPCE	Label-free format using Ab immobilized on AuMRs/PdNPs/PANI nanocomposite	DPV	TPM/shrimp-free cream crackers	0.01–100 pg mL^−1^	0.01 pg mL^−1^	84.1–117.6	-	[98]
GCE	Label-free format using Ab-oriented immobilization approach including SPA and AuNPs/PEI_MWCNTs nanocomposite	DPV	KBL/raw and cooked kidney bean milks	0.05–100 μg mL^−1^	0.023 μg mL^−1^	90.96–97.18	ELISA	[100]
SPCE	Sandwich format involving primary Ab immobilized on MBs and HRP-labeled secondary Ab	Amperometry	Sin a 1/raw plant extracts	2.7–50 ng mL^−1^	0.82 ng mL^−1^		ELISA	[104]
SPCE/SPdCE	Sandwich format involving primary Ab immobilized on MBs and HRP-labeled secondary Ab	Amperometry	β-conglycinin and glycinin/raw cookie dough and baked cookies enriched with soy flour	β-conglycinin 0.1–125 ng mL^−1^ glycinin 0.1–100 ng mL^−1^	β-conglycinin 0.03 ng mL^−1^ glycinin 0.02 ng mL^−1^	β-conglycinin 93–99% glycinin 101%	ELISA	[106]
AuSPE	Sandwich format involving primary Ab immobilized on MBs and HRPlabeled secondary Ab	Chronoamperometry	Gliadin, Ara h 1, Cor a 1; casein, OVA/bread, milk, cereal, cookies, ice cream, burgers, beers, dressed salads	-	Gliadin 0.075 mg kg^−1^ Ara h 1 0.007 mg kg^−1^ Cor a 1 0.089 mg kg^−1^ Casein 0.170 mg kg^−1^ OVA 0.003 mg kg^−1^	-	ELISA	[108]

*Abbreviations*: Ab: antibody; AuMRs: gold-microrods; AuSPE: gold screen-printed electrode; AuNCs: gold nanoclusters; AuNPs: gold nanoparticles; CGE: glassy carbon electrode; CHI: chitosan; CNFs: carbon nanofibers; CPE: carbon paste electrode; DPV: differential pulse voltammetry; EIS: electrochemical impedance spectroscopy; ELISA: enzyme-linked immunosorbent assay; G graphene; GNs: graphene nanosheets; rGO: reduced graphene oxide; GO: graphene oxide; KBL: kidney beans lectin; HRP: horse radish peroxidase; α-LB: α-lactoglobulin; β-LB: β-lactoglobulin; LOD: limit of detection; MBs: magnetic beads; MNPs: magnetic nanoparticles; MWCNTs: multi-walled carbon nanotubes; NDs: nanodiamonds; OM: ovomucoid; OVA: ovalbumin; PdNPs: palladium nanoparticles; PEI: polyethyleneimine; PPY: polypyrrole; S-AP: streptavidin-alkaline phosphatase; SAM: self-assembling monolayer; SPA: staphylococcal protein A; SPCE: screen-printed carbon electrode; SPdCE: screen-printed dual carbon electrode; SPCNTE: screen-printed carbon nanotube electrode; SPGE: screen-printed graphene electrode; SWV: square-wave voltammetry; TPM: tropomyosin.

**Table 2 biosensors-12-00503-t002:** Performance of electrochemical aptasensors for allergen detection.

Electrode	Immunosensor Format	Electrochemical Technique	Analyte/Sample	Linearity Range	LOD	Recovery (%)	Reference Method	Ref.
SPGE	Label-free format using aptamer immobilized onto SPGE and [Fe(CN)_6_]^4^^−/3^^−^ as the redox probe	SWV	β-LB/cake, cheese crackers, biscuits	100 pg mL^−1^–100 ng mL^−1^	20 pg mL^−1^	90–95	-	[111]
SPGE	Competitive format based on aptamer immobilization in PANI/PAA copolymer	DPV	β-LB/soy and cow milk	0.01–10 μg L^−1^	0.053 μg L^−1^	80–85 (soy) 95 (cow)	-	[113]
SPGE	Label-free format using an immobilized aptamer on AuNPs/poly(lysine) nanocomposite and MB as the redox probe	DPV	β-LB/biscuits, yogurt	0.1–10 ng mL^−1^	0.09 ng mL^−1^	103–117 (biscuits) 95–116 (yogurt)	-	[114]
ITOE	Aptasensor based on a highly selective DNA aptamer and flower-like Au@BiVO_4_ microspheres	Amperometry	β-LB/infant food formula	0.01–1000 ng mL^−^^1^	0.007 ng mL^−^^1^	92.0–103.5	ELISA	[117]
AuE	Aptasensor based on trifunctional HP, using AuNps and an HCR system	DPV	β-LB/hypoallergenic formula milk	0.01–100 ng mL^−^^1^	5.7 ng mL^−^^1^	94.5–101.4	ELISA	[124]
LSGE	Label-free format aptamer immobilization on CDI-CuNFs nanocomposite	ECS	β-LB/Herbalife meal replacement shake Formula 1	1 ag mL^−^^1^–100 fg mL^−^^1^	1 ag mL^−^^1^	92.95–94.99	-	[121]
SPCE	Competitive format involving HRP as the label	Chronoamperometry	Gliadin/gluten-free snacks and foods, rolled oats	1–100 μg mL^−1^	0.113 μg mL^−1^		ELISA	[124]
AuE	Label-free format including PMAMG4 as the immobilization layer	EIS	Gliadin/beer, gluten-free beer, rice, gluten-free bread, corn flour	5–50 mg mL^−1^50–1000 mg mL^−1^	5 mg mL^−1^	-	ELISA	[126]
SPCE	Label-free format involving AuNPs and streptavidin layer for aptamer immobilization	EIS	Gliadin/gluten-free beers, gluten-free soy sauce	0.1–1 mg L^−1^	0.05 mg L^−1^	93–101	ELISA	[130]
SPCE	Sandwich format involving two biotinylated aptamers and HRP as the enzymatic label	Chronoamperometry	Gliadin/dessert powders, panna cotta, vanilla cream	1–100 μg mL^−1^	1 μg mL^−1^	-	ELISA	[136]
SPCE	Sandwich format involving aptamer/Ab sandwich and HRP as the enzymatic label	Chronoamperometry	Gliadin/gluten-free flour, corn flakes	0.2–20 mg L^−1^	0.2 mg L^−1^	-	ELISA	[137]
SPCNTE	Label-free format using a printable ink including a CNT–aptamer complex and [Fe(CN)_6_]^4−/3−^ as the redox probe	EIS	Lys/-	0–1.0 μg mL^−1^	90 ng mL^−1^	-	-	[140]
GCE	Label-free format using an rGO/MWCNTs/CQDs/CHI nanocomposite	DPV/EIS	Lys/egg white, wine	20 fmol L^−1^–10 nmol L^−1^ (DPV) 10 fmol L^−1^–100 nmol L^−1^ (EIS)	3.7 fmol L^−1^ (DPV) 1.9 fmol L^−1^ (EIS)	94.6–96.0 (wine) 96.0–104.0 (egg)	-	[141]
SPCE	Label-free format using a NH_2_-rGO/IL/Nh2-MSNPs nanocomposite	DPV/EIS	Lys/egg white, wine	10 fmol L^−1^–50 nmol L^−1^ (DPV) 10 fmol L^−1^–200 nmol L^−1^ (EIS)	4.2 fmol L^−1^ (DPV) 2.1 fmol L^−1^ (EIS)	94.0–96.2 (wine) 95.4–104.2 (egg)	-	[142]
AuSPE	Label-free format involving AuNP-modified electrode and [Fe(CN)_6_]^4−/3−^ as the redox probe	CV	Lys/red and white wines	1–10 μg.mL^−^^1^	0.32 μg.mL^−^^1^	-	HPLC	[143]
SPCE	Sandwich format involving a thiolated aptamer and a secondary aptamer labeled with S-AP	DPV	Lys/red white and rose wines	70–7 × 10^5^ pM	2 pM	97.4–109.7	Qubit^®^ Fluorescence Protein Assay Kit	[144]
SPCE	Label-free format including microfluidic origami nano-aptasensor and BPNSs	DPV	Ara h 1/cookie dough	50–1000 ng mL^−1^	21.6 ng mL^−1^	98.3–107.9	-	[145]

*Abbreviations*: Ab: antibody; AuSPE: gold screen-printed electrode; AuNCs: gold nanoclusters; AuNPs: gold nanoparticles; BPNSs: black phosphorous nanosheets; BSA: bovine serum albumin; CDI: 1,1-carbonyldiimidazole; CGE: glassy carbon electrode; CHI: chitosan; CNT: carbon nanotube; CQDs: carbon quantum dots; DPV: differential pulse voltammetry; ECS: electrochemical capacitance spectroscopy; EIS: electrochemical impedance spectroscopy; ELISA: enzyme-linked immunosorbent assay; G graphene; rGO: reduced graphene oxide; GO: graphene oxide; HCR: hybridization chain reaction; HP: hairpin; HRP: horse radish peroxidase; IL: ionic liquid; ITO: indium tin oxide; β-LB: β-lactoglobulin; LOD: limit of detection; LSG: laser scribed graphene; Lys: lysozyme; MB: methylene blue; MWCNTs: multi-walled carbon nanotubes; NFs: nanoflowers; Nh2-MSNs; amino-mesosilica nanoparticles; PAA: poly(anthranilic acid); PAMAMG4: poly(amidoamine) dendrimer of fourth generation; PANI: ply(aniline); S-AP: streptavidin-alkaline phosphatase; SPCE: screen-printed carbon electrode; SPCNTE: screen-printed carbon nanotube electrode; S-AP: SPGE: screen-printed graphite electrode; SWV: square-wave voltammetry.

**Table 3 biosensors-12-00503-t003:** Performance of electrochemical genosensors, cell-based and MIP-based sensors for allergen detection.

Electrode	Biosensor Type and Format	Electrochemical Technique	Analyte/Sample	Linearity Range	LOD	Recovery (%)	Reference Method	Ref.
SPCE	Genosensor with sandwich format using MBs and HRP as enzymatic labels	Amperometry	Cor a 9/hazelnut, nuts, and fruit	0.0024–0.75 nM	0.72 pM	-	-	[150]
SPCE	Genosensor with sandwich format using MBs	Amperometry	Sola l 7/tomato, corn	0.8–50 pM	0.2 pM			[152]
SPCE	Cell-based biosensor using RBL-2H3 cells immobilized on CNFs/GelMA nanocomposite	DPV	Casein/-	1 × 10^−7^–1 × 10^−6^ g mL^−1^	3.2 × 10^−8^ g mL^−1^	-	-	[159]
SPCE	Cell-based biosensor based on a 3D paper chip using RBL-2H3 cells immobilized on PGHAP composite hydrogel	Capacitance	Ara h 2/raw and fried peanuts	0.1–1 ng mL^−1^	0.028 ng mL^−1^	-	-	[162]
SPCE	Cell-based biosensor based on RBL-2H3 cells immobilized on a biomimetic intestinal microvillus made with a bioink including FCONPs, MWCNTs-CDH, and GelMA	EIS	Gliadin/gluten-free flour and cookies	0.1–0.8 ng mL^−1^	0.036 ng mL^−1^	95.4–105.0	-	[163]
AuE	Bacteriophage-based biosensor using M13 phage immobilized on the electrode surface	SWV	OM/ egg, white wine	1.55–12.38 μg mL^−1^	0.12 μg mL	97.5–108.0 (egg white) 97.2–103.8 (wine)	-	[171]
GCE	MIP sensor including CHI and PPY as MIP	DEIS	BSA/human blood serum	0.0001–1 ng mL^−1^	5 × 10^−5^ ng mL^−1^	98–102	HPLC	[177]
SPCE	MIP sensor including choline chloride as the functional monomer and PEI-rGO-AuNCs as the nanocomposite	DPV	β-LB/milk	10^−^^9^–10^−^^4^ mg mL^−1^	10^−^^9^ mg mL^−1^	-	ELISA	[178]
AuSPE	MIP sensor including PPY as MIP	SWV	Cor a 14/hazelnut present in pasta	100 fg mL^−1^–0.1 mg mL^−1^	24.5 fg mL^−1^	-	-	[179]
SPCE	MIP sensor including poly (o-PD) as MIP	DPV	Genistein/soymilk, cookies, soy sauce, hummus, salad dressings, gingerbread, and muffin	100 ppb–10 ppm	100 ppb	-	LF	[180]
CPE	MIP sensor including SPIONs and PMMA as MIP	Amperometry	Gliadin/gluten-free and not gluten-free crackers	50–1000 ppm	1.50 ppm	-	-	[181]

*Abbreviations*: Ab: antibody; AuE: gold electrode; AuNCs: gold nanoclusters; BSA: bovine serum albumin; CGE: glassy carbon electrode; CHI: chitosan; CNFs: carbon nanofibers; CPE: carbon plate electrode; DPV: differential pulse voltammetry; DEIS: dynamic electrochemical impedance spectroscopy; EIS: electrochemical impedance spectroscopy; FCONPs: flower-like copper nanoparticles; GelMA: gelatin methacryloyl; HRP: horse radish peroxidase; LOD: limit of detection; MBs: magnetic beads; MIP: molecularly imprinted polymer; MWCNTs: multi-walled carbon nanotubes; MWCNTs-CDH: hydrazide functionalized multi-walled carbon nanotubes; o-PD: o-phenylenediamine; PGHP: polyvinyl alcohol gelatin methacryloyl nano-hydroxyapatite; PMMA: poly (methyl methacrylate); PPY: poly (pyrrole); RBL: rat basophilic leukemia; SPCE: screen-printed carbon electrode; SPIONs: superparamagnetic iron oxide nanoparticles; SWV: square-wave voltammetry.

## Data Availability

Not applicable.

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
