# Peer review of "Recent Advances in Electrochemical Sensing Strategies for Food Allergen Detection"

_biosensors, 2022, doi:10.3390/bios12070503_

Round 1

Reviewer 1 Report

The review article is very comprehensive, containing high quality information on sensors, specifically on electrochemical strategies to detect different allergens in food using different biological systems. In addition, the author mentions relevant aspects on manufacturing and values compared to conventional methods. However, some fragments of the text could be omitted to avoid being repetitive, such as the last part of the introduction, since that information is mentioned in the abstract and presented in the table of contents as well.

There are some punctuation errors (line 292 and others) in the text, some capitalization in the word Section (line 125). Some grammatical issues need to be revised in the plural words (line 168), some words are inappropriately cut by hyphens (line 179 and others), perhaps modifying the writing style could improve those errors.

There is an irrelevant sentence in line 238-239.

In line 279-280 the Faradian approaches are mentioned first but then the non-Faradian approaches are explained first, I will recommend to follow the order of those mentioned to avoid jumps in the subtopics and to facilitate the reading.

There is a sentence in line 313-315 that is irrelevant, as the audience of the article will look for more information if they deem it necessary.

The sentence of line 337-338 is repetitive

Author Response

The response is included in the dile attached

Reviewer 2 Report

The review “Recent advances in electrochemical sensing strategies for food allergens detection” has an interesting topic and it is well written. This way, I recommend the publication of this review in Biosensors. I have comments and suggestions to enhance the performance of review, enumerated below.

1.     In section 3 the author has define biosensor according to ref 22 (Biosensors and Bioelectronics, 2001). A novel reference with the IUPAC definitions is available. See the IUPAC recommendations (Pure Appl. Chem. 2020; 92(4): 641–694; item 4.6) and verify the ‘biosensor’ and other definitions that were reformulated.

2.     In line 249, the author says that ‘analyte is reduced’. But it can be oxidized too using chronoamperometric technique. Correct this information.

3.     I don’t agree with the information in line 256. Voltammetry is a technique that vary the potential and measure the generated current in the working electrode. Amperometry uses a constant potential in function of time.

4.     The information in lines 276-278 is vague. Explain.

5.     The review is based on “Recent advances” in electrochemical sensing, but less than 10% of references are from 2022. There are several works from 2022 that can be added in the review.

6.     I missed more Figures and Schemes in the whole review.

7. The english must be revised.

Author Response

The response is included in the file attached.

Round 2

Reviewer 2 Report

The author have been enhance the quality of the work with the suggestions, therefore I recommend the publication in Biosensors.